# Targeted Neural Dynamical Modeling

**Cole Hurwitz**
School of Informatics
University of Edinburgh
Edinburgh, Scotland, EH8 9AB
colehurwitz@gmail.com

**Akash Srivastava**
MIT-IBM Watson AI Lab
Cambridge, MA 02142,
Akash.Srivastava@ibm.com

**Kai Xu**
School of Informatics
University of Edinburgh
Edinburgh, Scotland, EH8 9AB
xukai921110@gmail.com

**Justin Jude**
School of Informatics
University of Edinburgh
Edinburgh, Scotland, EH8 9AB
justin.jude@ed.ac.uk

**Matthew G. Perich**
Icahn School of Medicine at Mount Sinai
New York, NY 10029
mperich@gmail.com

**Lee E. Miller**
Feinberg School of Medicine
Northwestern
Chicago, IL 60611
lm@northwestern.edu

**Matthias H. Hennig**
School of Informatics
University of Edinburgh
Edinburgh, Scotland, EH8 9AB
m.hennig@ed.ac.uk

## Abstract

Latent dynamics models have emerged as powerful tools for modeling and interpreting neural population activity. Recently, there has been a focus on incorporating simultaneously measured behaviour into these models to further disentangle sources of neural variability in their latent space. These approaches, however, are limited in their ability to capture the underlying neural dynamics (e.g. linear) and in their ability to relate the learned dynamics back to the observed behaviour (e.g. no time lag). To this end, we introduce Targeted Neural Dynamical Modeling (TNDM), a nonlinear state-space model that jointly models the neural activity and external behavioural variables. TNDM decomposes neural dynamics into behaviourally relevant and behaviourally irrelevant dynamics; the relevant dynamics are used to reconstruct the behaviour through a flexible linear decoder and both sets of dynamics are used to reconstruct the neural activity through a linear decoder with no time lag. We implement TNDM as a sequential variational autoencoder and validate it on simulated recordings and recordings taken from the premotor and motor cortex of a monkey performing a center-out reaching task. We show that TNDM is able to learn low-dimensional latent dynamics that are highly predictive of behaviour without sacrificing its fit to the neural data.

# 1   Introduction

Recent progress in high-density, microelectrode array technology now allows for recording from hundreds to thousands of neurons with the precision of single spikes [11]. Despite the apparent high dimensionality of these datasets, neural activity is often surprisingly well-explained by low-dimensional latent dynamics [4, 24, 6, 8]. Extracting these dynamics from single trials is crucial for understanding how neural activity relates to a behavioural task or stimulus [17].

Latent variable models (LVMs) are a natural choice for capturing low-dimensional structure from neural activity as they can learn to map a few latent variables to arbitrarily complicated response structure in the activity. Already, there exist a number of LVMs that have been successfully applied to neural data ranging from simple non-temporal models such as principal components analysis (PCA) [5] to complex state-space models such as LFADS [17]. In these models, the goal is to learn a set of latent factors that best explain neural variability. As such, there is no guarantee that the different sources of variability present in the population activity will be disentangled in the latent space (e.g. behaviour, arousal, thirst, etc.) [25, 10].

To better partition sources of neural variability in the latent space, some LVMs have been developed that incorporate an external behaviour into the generative process [14, 19, 31]. These methods, however, do not model temporal dependencies between the latent states. Recently, a novel state-space model termed preferential subspace identification (PSID) was developed that jointly models neural activity and behaviour with a shared set of dynamics [25]. When applied to neural activity recorded in the premotor cortex (PMd) and primary motor cortex (M1) of a monkey during a 3D reaching task, PSID was shown to extract latent factors that were more predictive of behaviour than the factors extracted by other approaches. Despite the strength and simplicity of this approach, it suffers from two main drawbacks. First, PSID is a linear state-space model and cannot capture the nonlinear dynamics which are thought to underlie phenomena such as rhythmic motor patterns [22, 9] or decision making [21]. Second, PSID assumes that behaviourally relevant dynamics explain both the neural activity and behaviour with no time lag. This limits the ability of PSID to capture more complex temporal relationships between the latent dynamics and the behaviour.

In this work, we introduce Targeted Neural Dynamical Modeling (TNDM), a nonlinear state-space model that jointly models neural activity and behaviour. Similarly to PSID, TNDM decomposes neural activity into behaviourally relevant and behaviourally irrelevant dynamics and uses the relevant dynamics to reconstruct the behaviour and both sets of dynamics to reconstruct the neural activity. Unlike PSID, TNDM does not constrain the latent dynamics at each time step to explain behaviour at each time step and instead allows for any linear relationship (constrained to be causal in time) between the relevant dynamics and the behaviour of interest. We further encourage partitioning of the latent dynamics by imposing a disentanglement penalty on the distributions of the initial conditions of the relevant and irrelevant dynamics. To perform efficient inference of the underlying nonlinear dynamics, TNDM is implemented as a sequential variational autoencoder (VAE) [13, 29][1]. We compare TNDM to PSID and to LFADS, a nonlinear state-space model that only models neural activity, to illustrate that TNDM extracts more behaviourally relevant dynamics without sacrificing its fit to the neural data. We validate TNDM on simulated recordings and neural population recordings taken from the premotor and motor cortex of a monkey during a center-out reaching task. In this analysis, we find that the behaviourally relevant dynamics revealed by TNDM are lower dimensional than those of other methods while being more predictive of behaviour.

# 2   Background/Related work

**Notatation.**   Let $x \in \mathbb{N}^{N \times T}$ be the observed spike counts and let $y \in \mathbb{R}^{B \times T}$ be the observed behaviour during a single-trial.[2] We define the unobserved latent factors in a single trial as $z \in \mathbb{R}^{M \times T}$ where $M < N$. For TNDM, as with PSID, it is important to distinguish between behaviourally relevant $z_r$ and behaviourally irrelevant $z_i$ latent factors. The behaviourally relevant latent factors

---

[1]The code for running and evaluating TNDM on real data can be found at https://github.com/HennigLab/tndm_paper. We also provide a Tensorflow2 re-implemention of TNDM at https://github.com/HennigLab/tndm. It is important to note that all reportraed results for the *real* datasets use the old model and not the re-implementation. For the *synthetic* dataset results, we use the re-implementation.

[2]In this work, we assume that behaviour is temporal and has the same time length as recorded neural activity (e.g. hand position). TNDM can be extended to discrete/non-temporal behaviours (e.g. reach direction).

$z_r$ summarize the variability in the neural activity associated with the observed behaviour while the behaviourally irrelevant latent factors $z_i$ explain everything else in the neural data (Figure 1a). We assume that each of the unobserved, single-trial factors can be partitioned into these relevant and irrelevant factors $z := \{z_r, z_i\}$.

**State-space models for neural data.** There are a number of state-space models that have been developed and applied to neural population activity. The expressivity of these models range from simple linear dynamical models [27, 2, 20] to more complex nonlinear models where the latent dynamics are parameterized by recurrent neural networks (RNNs) [17, 26]. For this work, there are two state-space models that are most relevant: LFADS and PSID.

LFADS, or latent factor analysis via dynamical systems, is a state-of-the-art nonlinear state-space model for neural data. In LFADS, the latent dynamics are generated by sampling high-dimensional initial conditions $g_0$ from some distribution $p_{g_0}$ and then evolving $g_0$ with a deterministic RNN $f_\theta$. A linear mapping $W_z$ is then applied to the high-dimensional dynamics $g_t$ to transform them into the low-dimensional 'dynamical' factors $z$. These dynamical factors are transformed into spike counts by mapping each time point to a rate parameter $r$ of a Poisson distribution using a weight matrix $W_r$ followed by an exponential nonlinearity. The generative process is defined as: $g_0 \sim p_{g_0}, g_t = f_\theta(g_{t-1}), z_t = W_z(g_t), r_t = \exp(W_r(z_t)), x_t \sim \text{Poisson}(x_t|r_t)$. The initial conditions $g_0$ are inferred from $x$ with an RNN encoder network $q_\phi$. Utilizing the reparameterization trick, the model is trained using gradient descent and by optimizing the evidence lower-bound (ELBO) of the marginal log-likelihood. While LFADS provides an excellent fit to the neural data, it inevitably mixes different sources of neural variability in the latent dynamics $z$ as there is no constraints imposed to disentangle these dynamics.

PSID is a linear dynamical model that partitions the latent dynamics into behaviourally relevant and behaviourally irrelevant dynamics $z := \{z_i, z_r\}$. The dynamical evolution of $z$ is defined by a transition matrix $A$ along with a Gaussian noise term $w_z$. $z$ is transformed into the observed firing rates $x$ by mapping each time point to the mean of a Gaussian distribution with a weight matrix $W_x$ and noise term $w_x$. The behaviourally relevant dynamics $z_r$ at each time point are transformed into the observed behaviour $y$ with a weight matrix $W_y$. The state space model for PSID is then defined as: $z_t \sim \mathcal{N}(z_t|A(z_{t-1}), w_z), x_t \sim \mathcal{N}(x|W_x(z_t), w_x), y_t = W_y(z_r)$. PSID uses a novel two-stage subspace identification approach to learn the parameters of their model. In the first stage, PSID extracts the behaviourally relevant dynamics through an orthogonal projection of future behaviour onto past neural activity. The irrelevant dynamics are then extracted through an additional orthogonal projection of residual neural activity onto past neural activity. In comparison to LFADS, PSID was shown to extract latent states that are better able to predict behaviour when using a Kalman filter. Despite the analytical simplicity of this approach, it suffers from a few drawbacks. First, it can only model linear dynamics which may not provide a good fit to nonlinear activity patterns or behaviours (e.g. multi-directional reaches). Second, the relevant dynamics at each time step $z_{rt}$ must be mapped one-to-one to the behaviour $y_t$ during training (i.e. no time lag). This imposes a strong structural constraint on the relevant dynamics which hampers their ability to explain neural variability.

## 3 Model

In this work, we introduce Targeted Neural Dynamical Modeling (TNDM). TNDM is a nonlinear state-space model which jointly models neural activity and an observed behaviour. Crucially, TNDM learns to reconstruct both the population activity and behaviour by disentangling the behaviourally relevant and behaviourally irrelevant dynamics that underlie the neural activity.

**Generative model.** A plate diagram of TNDM's graphical model is shown in Figure 1a. We assume that the observed neural activity $x$ and behaviour $y$ in each trial are generated by two sets of latent factors $z_i$ and $z_r$.

The generative process of TNDM is defined below:

$$
\begin{aligned}
&g_{i0} \sim p_{g_{i_0}}, g_{r0} \sim p_{g_{r_0}}, \ g_{it} = f_{\theta_i}(g_{it-1}), \ g_{rt} = f_{\theta_r}(g_{rt-1}) \\
&z_{it} = W_{iz}(g_{it}), \ z_{rt} = W_{rz}(g_{rt}), \ r_t = \exp(W_r(z_{it}, z_{rt}))) \\
&x_t \sim \text{Poisson}(x_t|r_t), \ y \sim \mathcal{N}(y|C_y(z_r), I)
\end{aligned}
\tag{1}
$$

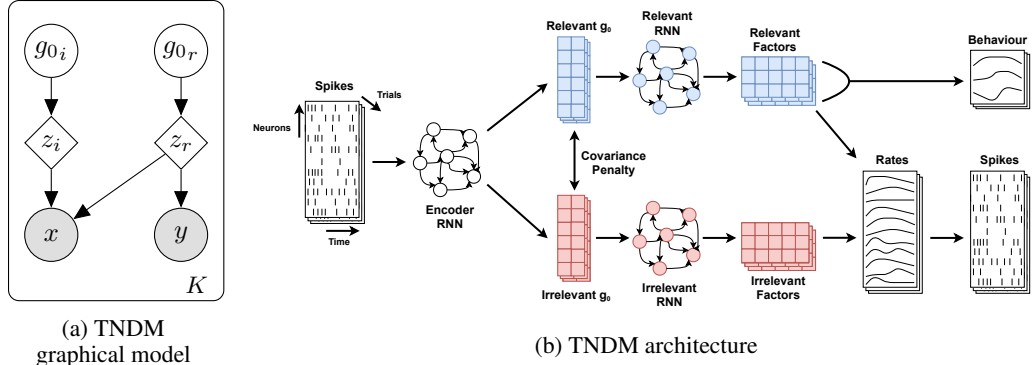

(a) TNDM
graphical model

(b) TNDM architecture

Figure 1: (a) The latent space of TNDM is partitioned into irrelevant and relevant high-dimensional initial conditions $g_{0_i}$ and $g_{0_r}$. These initial conditions are deterministically transformed to recover the latent factors $z_i$ and $z_r$ which give rise to the jointly observed neural activity $x$ and behaviour $y$. We assume there are K trials in the dataset. (b) TNDM utilizes a sequential variational autoencoding approach to ammortize inference of the relevant and irrelevant initial conditions $g_{r0}$ and $g_{i0}$. The initial conditions are passed through two separate RNNs to generate the behaviourally relevant and irrelevant dynamics $g_i$ and $g_r$ which are then projected into a low-dimensional subspace to recover the dynamical factors $z_i$ and $z_r$. These factors are used to reconstruct the neural activity and behaviour. The behaviour is reconstructed from the relevant factors using a flexible linear decoder which can capture complex temporal relationships (see the paragraph on **Behaviour Decoding**).

In the above equation, $p_{g_{i_0}}$ and $p_{g_{r_0}}$ are the distributions over the initial conditions of the behaviourally irrelevant and behaviourally relevant dynamics, respectively (assumed to be Gaussian). Similarly to LFADS, we parameterize the nonlinear transition functions $f_{\theta_i}$ and $f_{\theta_r}$ using RNNs[3]. As $g_{i_t}$ and $g_{r_t}$ can have arbitrarily high-dimensionality in our model (defined by the number of units in the RNN), we utilize two weight matrices, $W_{iz}$ and $W_{rz}$, to project these high-dimensional dynamics into a low-dimensional subspace, giving rise to the relevant and irrelevant dynamical factors $z_{i_t}$ and $z_{r_t}$ at each time step. These factors are then used to reconstruct both the observed neural activity and behaviour using linear decoders. An essential feature of our generative model is that although neural activity is reconstructed from the latent dynamics at each time step (i.e. no time lag), we let the relevant factors reconstruct the behaviour with a more flexible linear decoder $C_y$ that allows for time lags (explained in the **Behaviour Decoding** paragraph). We fix the variance of the Gaussian observation model for the behaviour (i.e. we utilize mean squared error) as naively inferring the variance in the decoder can lead to numerical instabilities [23].

It is important to understand that although the dimensionality of the dynamics in TNDM (and LFADS) can be arbitrarily high, the dimensionality of the subspace that gives rise to neural activity and behaviour will be low due to the projection. Therefore, our model can be used to examine the number of latent variables (i.e. activity patterns) that are needed to characterize the population response and corresponding behaviour. As this is the primary goal when fitting LVMs to neural data [5], we compare all LVMs in this paper (TNDM, LFADS, and PSID) by the dimensionality of this subspace rather than the dimensionality of the dynamics.

**Behaviour Decoding.** As mentioned above, PSID utilizes a linear weight matrix that maps the relevant latent dynamics at each time step to the behaviour at each time step, i.e $y_t = W_y(z_{r_t})$. This parameterization does not allow for modeling any latency, long-term dependencies or correlations, therefore, it severely limits the ability of the relevant dynamics to simultaneously explain neural activity and behaviour. To demonstrate the drawbacks of the no time lag behaviour decoder, we show that while training TNDM using this decoder leads to accurate behaviour prediction, the reconstruction of the neural activity noticeably decreases. This issue gets exacerbated in models with nonlinear dynamics as the expressivity of the underlying RNNs, along with the inflexible one-to-one behaviour mapping, leads the relevant dynamics to simply learn to replicate the behaviour of interest. These results are summarized in Supplement 1.

---

[3]To implement TNDM, we primarily adapt the original Tensorflow [1] implementation of LFADS from https://github.com/lfads/models (Apache License 2.0).

To overcome this limitation we instead allow the relevant latent dynamics to reconstruct the behaviour through any learned linear causal relationship. To this end, we introduce a linear weight matrix $C_y$ with dimensionality $n_{z_r}T \times BT$ where $n_{z_r}$ is the number of relevant dynamics, $B$ is the number of behaviour dimensions, and $T$ is the time length of a single trial. To transform the relevant factors $z_r$ into behaviours using $C_y$, we concatenate each dimension of $z_r$ in time to form a 1D vector $Z_r$ with length $n_{z_r}T$ and then perform the operation $Y = C_y Z_r$ where Y is the resulting concatenated behaviour. As $Y$ is a 1D vector with length $BT$, we can reshape $Y$ to recover the reconstructed behaviour $\hat{y}$. Importantly, we do not allow acausal connections in $C_y$, i.e. the lower triangular components of each of the dynamics to behaviour blocks are set to zero during training. For an example weight matrix $C_y$, see Figure 4a. In comparison to a simple no time lag mapping, we find that our flexible, causal linear decoder allows the relevant latent dynamics to both reconstruct the measured behaviour and capture neural variability. This is shown in Supplement 1 where the behaviourally relevant factors learned by TNDM with the full causal decoder contribute more meaningfully to the neural reconstruction than when using the no time lag decoder.

**Inference**   To extract the latent dynamics $z_r$ and $z_i$ from the neural activity $x$, we first learn to approximate the posterior over the initial conditions of the dynamics $g_{r0}$ and $g_{i0}$. Then, we learn a deterministic mapping from the initial conditions to the latent dynamics $z_r$ and $z_i$. To approximate the true posterior $p(g_{r0}, g_{i0}|x, y)$, we implement TNDM as a sequential VAE and define our variational posterior as the product of two independent multivariate Gaussian distributions with diagonal covariances. The variational parameters of each Gaussian are computed with a shared encoder network $e_{\phi_0}$ followed by separate linear transformations. This shared encoder is implemented as a bi-directional RNN such that the initial conditions are inferred using information from the full trial. The equation for inferring the variational posterior is shown below:

$$q_{\Phi_r}(g_{r0}|x)q_{\Phi_i}(g_{i0}|x) = \mathcal{N}(\mu_{\phi_{r_1}}(e_{\phi_0}(x)), \sigma^2_{\phi_{r_2}}(e_{\phi_0}(x))) \cdot \mathcal{N}(\mu_{\phi_{i_1}}(e_{\phi_0}(x)), \sigma^2_{\phi_{i_2}}(e_{\phi_0}(x)))$$
(2)

The inference networks for the behaviourally relevant and the behaviourally irrelevant initial conditions are parameterized by $\Phi_r = \{\phi_0, \phi_{r_1}, \phi_{r_2}\}$ and $\Phi_i = \{\phi_0, \phi_{i_1}, \phi_{i_2}\}$ where $\phi_0$ are the parameters of the shared RNN. It is important to note that TNDM's variational posterior only depends on the neural activity $x$. This approximation forces the learned initial conditions to come from the observed activity and allows for decoding of unseen behaviours after training. The reparameterization trick is used to sample from each initial condition distribution and the sampled initial conditions are evolved using separate decoder RNNs to produce the behaviourally relevant $g_{r_t}$ and irrelevant high-dimensional dynamics $g_{i_t}$. The high-dimensional dynamics at each time-step are projected into a low-dimensional subspace to recover the low-dimensional dynamical factors $z_{r_t}$ and $z_{i_t}$. The neural activity $x$ and behaviour $y$ are generated from the latent factors $z$ as shown in Equation 1.

The ELBO for the observed data from a single trial is therefore defined as:

$$\text{ELBO}(x, y) = -\text{KL}\left[q_{\Phi_r} \| p_{g_{r_0}}\right] - \text{KL}\left[q_{\Phi_i} \| p_{g_{i_0}}\right] + \mathbb{E}_{q_{\Phi_r}q_{\Phi_i}}[\log p_{\theta_1}(x|g_i, g_r)p_{\theta_2}(y|g_r)] \quad (3)$$

where $p_{\theta_1}$ and $p_{\theta_2}$ are the observation models for the neural activity and behaviour, respectively.

**Disentangling the latent dynamics**   Despite factorising the variational posterior, the true posterior over the latent variables $P(g_{0_i}, g_{0_r}|x, y)$ cannot be factorized; that is, $z_i$ and $z_r$ (which are deterministic transforms of $g_{0_i}$ and $g_{0_r}$) are conditionally dependent given the observed data. This means that $z_i$ and $z_r$ will be statistically dependent. To reduce sharing of information and redundancy between these two sets of dynamics, we introduce a novel disentanglement penalty on the two initial condition distributions. For this penalty, we take inspiration from the domain of unsupervised disentangled representation learning where it is standard to introduce additional penalties that encourage disentanglement or independence of the latent representations [15, 3, 12]. As mutual information is hard to compute for high-dimensional variables and the experimental data often has a very limited number of trials to estimate these distributions reliably, we instead penalize the mean of the sample cross-correlations between $g_{0_r}$ and $g_{0_i}$. Importantly, this cross-correlation penalty is applied in such a way that the final objective is still a valid lower bound of the log-likelihood (the cross-correlation

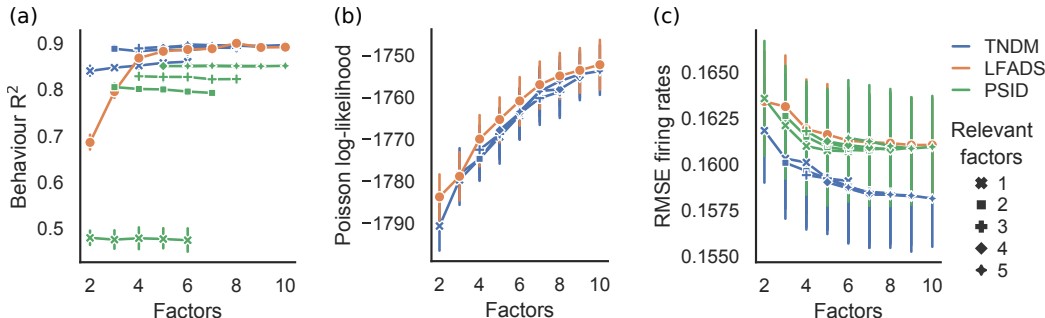

Figure 2: Summary of the behaviour and activity reconstruction accuracy for TNDM, LFADS, and PSID fit to neural recordings from the monkey primary motor cortex (M1) during a center-out reaching task. Each plot shows performance as a function of the total number of latent factors, averaged over five fits with different initialisation (random seeds) and different random training-test data splits. Error bars show standard error of the mean. (a) Coefficient of determination ($R^2$) for measured and reconstructed behaviour (hand position). For TNDM, the reconstruction performed by the behaviour decoder (with only the relevant factors) while for LFADS, a ridge regression had to be used to decode the behaviour ex post factor. For PSID the reconstruction by the model was additionally Kalman smoothed. (b) Poisson log-likelihood for the activity reconstruction per single trial for TNDM and LFADS. (c) Root mean square error (RMSE) between the predicted and actual ground-truth firing rates. Averaging was performed across all trials with the same movement direction. Behaviour reconstruction and log-likelihood were computed on held out test data, and the firing rate RMSE on the whole data set to allow for more reliable averaging.

penalty is always negative). We refer to this penalty as $Q(q_{\Phi_r}, q_{\Phi_i})$ and we adjust its weight with a hyperparameter $\lambda_Q$ (see Supplement 4 for an ablation study of this penalty). The final objective function for TNDM is then:

$$
\begin{aligned}
J(x,y) = -\,\mathrm{KL}\left[q_{\Phi_r} \,\|\, p_{g_{r_0}}\right] - \mathrm{KL}\left[q_{\Phi_i} \,\|\, p_{g_{i_0}}\right] + \mathbb{E}_{q_{\Phi_r} q_{\Phi_i}}[\log p_{\theta_1}(x|g_i, g_r)] + \\
\lambda_b \mathbb{E}_{q_{\Phi_r}}[\log p_{\theta_2}(y|g_r)] + \lambda_Q Q(q_{\Phi_r}, q_{\Phi_i})
\end{aligned}
\tag{4}
$$

where $\lambda_b$ is an additional hyperparameter introduced to balance the behavioural likelihood with the neural likelihood (see Supplement 3 for hyperparameter details). While TNDM is not the first VAE to jointly model two observed variables with a partitioned latent space [30], it is distinguished by its unique objective function and penalties, its RNN-based architecture, its causal linear decoder, and its novel application to neural activity and behaviour.

## 4 Experiments

### 4.1 Simulated Data

We evaluate TNDM on synthetic spike trains generated from a Lorenz system, a common benchmark for state space models of neural activity. For a detailed description of the simulated data and evaluation, we refer the reader to Appendix 6.

### 4.2 M1 neural recordings during reach

We apply TNDM to data gathered from a previously published monkey reaching experiment [7]. The monkey is trained to perform a center-out reaching task with eight outer targets. On a go cue, the monkey moves a manipulandum along a 2D plane to the presented target and receives a liquid reward upon success. Spiking activity from M1 and PMd along with the 2D hand position are recorded during each trial. We train each model on single-session data gathered from one of the six trained monkeys. The data consist of two paired datasets: PMd activity paired with hand position and M1 activity paired with hand position. We show results for the M1 recordings in the main text and the results for the PMd recordings in Supplement 2. The neural activity is counted in 10ms bins and the behaviour is also measured every 10ms. We align the behaviour to the spikes for both datasets

by taking the activity starting during movement onset. We set the length of the neural activity to be the minimum time until the target is reached across all trials. As one of our baselines, PSID cannot model spike count data, we smooth the spike counts with a Gaussian kernel smoother (with standard deviation 50ms) before applying PSID. Out of the 176 trials from the experiment, we use 80% for training (136 trials). We hold out the remaining 34 trials to test the models.

**Models/Evaluation** For all models, we perform a sweep over the number latent factors. For TNDM and PSID, we train models with all combinations of 1-5 relevant latent factors and 1-5 irrelevant factors (e.g. 3 relevant and 2 irrelevant). For LFADS, we train models with the number of latent factors ranging from 2-10. As TNDM and LFADS are both implemented as sequential variational autoencoders, we fix the architectures to be same for the two methods (64 units in the generators and encoder). We fix all shared hyperparameters to be the same between the two methods except for the dropout (TNDM requires more regularization due to the behavioural loss). For a detailed explanation of the hyperparameters and architectures used in these experiments, see Supplement 3. All results reported here are based on five fits of each model with different random seeds and data splits.

To compare TNDM to LFADS, we first evaluate their neural reconstruction using the test data Poisson log-likelihood and the root mean square error (RMSE) between the predicted and actual ground-truth firing rates. To calculate the ground-truth firing rates, we average the neural data across all trials with the same movement direction and used both the training and test sets to get more robust estimates of the rates from the experimental data. To evaluate the behaviour reconstruction of LFADS, we perform an ex post facto regression from the extracted latent factors to the behaviour in the training set. This regression is linear and is from all time steps of the factors to all time steps of the behaviour[4]. Note that this approach for regressing the LFADS factors is more flexible than the decoder in TNDM which is also linear but constrained to be causal. We then compute the coefficient of determination ($R^2$) between the decoded and ground-truth behaviour for each model on the test data.

To compare TNDM to PSID, we evaluate the neural reconstruction by computing the RMSE between the predicted and actual ground-truth firing rates. For behaviour reconstruction, we compute the $R^2$ between the decoded and ground-truth behaviours for each model on the test data.[5] As both TNDM and LFADS use information from the whole trial to infer the latent factors (which is inherently acausal), we use a standard Kalman smoother for PSID to make the state estimation comparable.

## 5 Results

### 5.1 Simulated Data

For a detailed discussion of TNDM's results on synthetic spike trains generated from a Lorenz system, we refer the reader to Appendix 6.

### 5.2 M1 neural recordings during reach

**Fit to behaviour** The behaviour reconstruction of TNDM, LFADS and PSID is summarized in Figure 2a. LFADS behavioural reconstruction saturates at around eight factors ($R^2 \approx 0.89$) and with just four factors yields a respectable behavioural fit ($R^2 \approx 0.86$). This indicates that the LFADS factors, which are constrained only by neural activity, are interpretable in terms of externally measured variables. In comparison, TNDM achieves saturating performance with just three latent factors ($R^2 \approx 0.90$) where *only two are constrained to be relevant* for behavioural reconstruction. In fact, all TNDM models with three or more factors (where at least two are constrained to be relevant) have similar behaviour reconstruction accuracy. In comparison to TNDM,

---

[4]We utilize a standard ridge regression from scikit-learn [18] with default parameters.

[5]A potential concern when comparing TNDM and PSID on behaviour reconstruction is that TNDM has more parameters in its behaviour decoder than PSID does. This is because TNDM decodes the behavioural variable at time step $t$ using all past time steps of the latent factors while PSID only uses the current time step $t$. As shown in Supplement 1, however, TNDM achieves equally high behaviour reconstruction using the no time lag decoder as it does using the proposed casual decoder, therefore, the number of parameters in TNDM's behaviour decoder is not a confounding factor for this evaluation. Also, while it is possible to remap PSID's learned latent factors to the behaviour using a higher parameter regression, this would be equivalent to changing PSID's generative model and, therefore, would no longer be a valid comparison to PSID.

LFADS achieves a behaviour reconstruction of just ($R^2 \approx 0.68$) for three latent factors. TNDM also has much more accurate behaviour reconstruction on the test data than PSID. For three latent factors, where two are constrained to be relevant, PSID achieves a behavioural fit of ($R^2 \approx 0.82$). PSID's behavioural reconstruction saturates at six latent factors where five are constrained to be relevant $R^2 \approx 0.88$). Overall, TNDM's behaviour decoding performance implies that the dimensionality of the behaviourally relevant dynamics for this 2D task are lower-dimensional than previously predicted by other latent variable modeling approaches.

**Fit to neural activity**   Do the additional constraints and penalties of TNDM affect the accuracy of neural activity reconstruction? This is an important question to answer as the learned latent dynamics are only meaningful if they also explain the observed activity well. Surprisingly, we find that TNDM's and LFAD's Poisson log-likelihoods on the test data are very close (Figure 2b). This indicates that the partitioning of the latent dynamics and the additional constraints imposed by TNDM have a very small effect on its neural reconstruction for this dataset. Instead, TNDM and LFADS both show a gradual improvement of neural activity reconstruction as a function of the number of factors. The only deviation from this trend is TNDM with one relevant and one irrelevant factor. This is not surprising, however, as much of the neural variability is explained by the behaviour of interest; only allowing one latent factor to explain the neural variability related to behaviour (while simultaneously enforcing disentanglement between the relevant and irrelevant dynamics) will cause TNDM's neural activity reconstruction to suffer. Perhaps more surprisingly, TNDM achieves a lower firing rate RMSE than LFADS with the same number of factors (Figure 2c). While on the surface this result seems counterintuitive, it may be because the RMSE is computed for the average firing rate over all trials of the same movement direction. While TNDM and LFADS have a very similar Poisson likelihood on single trials, TNDM can better distinguish trials by movement direction since it is explicitly modeling behaviour, hence the firing rate prediction split by trial type is improved. PSID provides a worse fit to the neural data than TNDM which is expected given that it is constrained to learn linear dynamics.

**PSID failure mode**   Although the neural reconstruction is fairly good for PSID, we find an unexpected result when analyzing PSID's learned model parameters. Specifically, we find that PSID's state-transition matrix $A$, which characterizes the behaviourally relevant neural dynamics [25], is approximately the identity matrix for this dataset and is non-informative about the neural activity. We expand upon this analysis of PSID in Supplement 5 where we show that PSID recovers the same state-transition matrix $A$ when we shuffle the neural data by trial or by time. We provide further evidence that PSID is unable to find informative linear dynamics for this dataset because the behaviour is inherently nonlinear across trials (i.e. multi-directional reaches). Therefore, on this dataset, we conclude that PSID's performance on neural reconstruction is mainly determined by the behaviourally irrelevant factors and its performance on behaviour reconstruction is completely determined by the Kalman gain during decoding.

**Interpretation of the learned dynamics**   In Figure 3, we visualize each stage of the generative process of TNDM (2 relevant and 2 irrelevant factors) and LFADS (4 factors) after training both models on the M1 dataset. As can be seen in the figure, there appears to be a clear separation between the relevant and irrelevant initial condition distributions in TNDM that is less apparent in the mixed latent space of LFADS. In fact, the relevant initial conditions of TNDM seem to precisely capture the reach direction of the behaviour. Despite the noticeable differences between the dynamics of LFADS and TNDM, their ability to infer the underlying firing rates from this dataset are nearly identical.

Looking at the learned dynamical factors for TNDM, one can see that the relevant dynamics are more clearly distinguished by reach condition and there is much less variance in the learned trajectories than those of LFADS. At the same time, the relevant TNDM factors do not trivially re-capitulate the behaviour dynamics, indicating that the dual constraint of behaviour and neural activity unmasks a more complicated relationship between the two. This relationship can be analysed by visualizing the learned weights of TNDM's behaviour decoder as shown in Figure 4a (for two relevant factors and two irrelevant factors). In this weight matrix, each time point of the behaviour receives contributions from a broad time interval of preceding factor activity. This corresponds to a temporal integration of the factors and suggests that the relevant factors represent information about movement velocity. Indeed, velocity can be decoded well from the relevant factors using a simple ridge regression (both for TNDM and LFADS, Figure 4c). The learned coefficients of this ridge regression for TNDM have a diagonally banded structure that corresponds to a delayed identity transformation (Figure 4b),

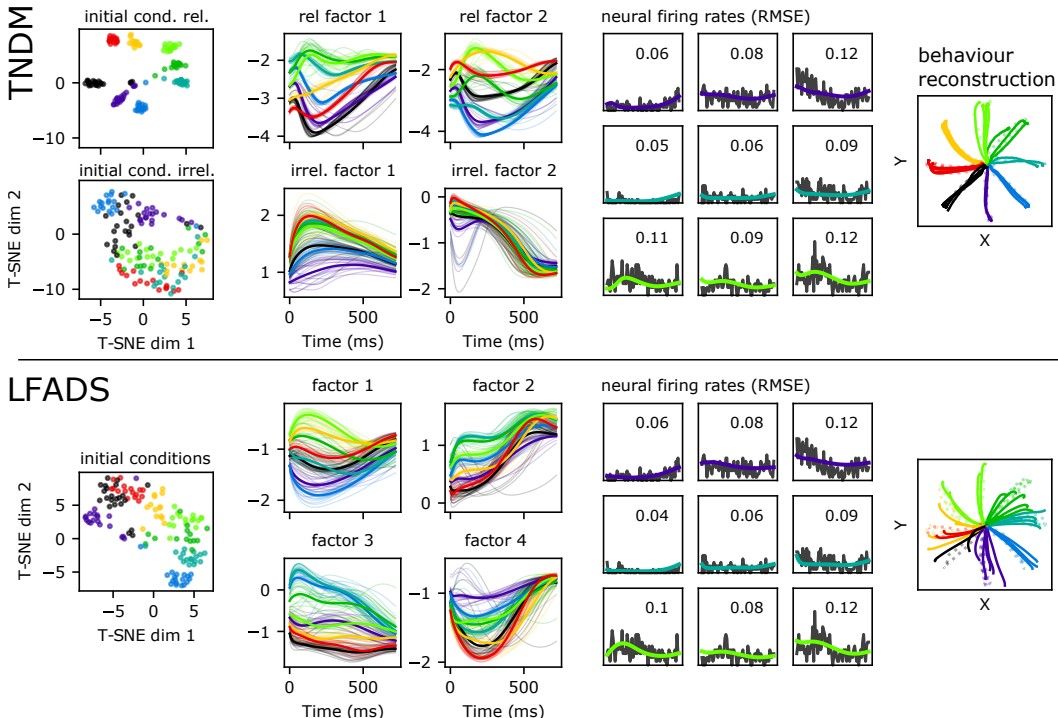

Figure 3: We visualize each component of the generative process for TNDM (top) and LFADS (bottom) after training. On the far left, we visualize the inferred initial conditions for each method after reducing the dimension to 2 with T-SNE. As can be seen, TNDM's inferred initial conditions show a clear distinction between behaviourally relevant and behaviourally irrelevant information whereas LFADS inferred initial conditions mix this information together. Next, we show the condition-averaged inferred latent dynamical factors (along with the single-trial factors) for each method to demonstrate that there is a clear distinction between the behaviourally relevant and behaviourally irrelevant factors in TNDM but not in LFADS. Finally, we show neural activity reconstruction (numbers are RMSE between data and prediction) and behaviour reconstructions (linear regression for LFADS) for both methods to illustrate that TNDM provides an excellent fit to the neural data despite the partitioned latent space and behavioural prediction.

which is not visible for the LFADS factors (not illustrated). Taken together, these results suggest that M1 neural dynamics are related to velocity of the hand in this task. Interestingly, we find that velocity decoding peaks at two relevant factors for TNDM and is less discernible when this number is increased, indicating that the addition of more relevant factors may spread this velocity information across the factors in a nonlinear way which cannot be recovered by the ridge regression (not visualized in Figure 4). It also illustrates that the TNDM's behaviour prediction saturation point (two relevant factors) has perhaps the most interpretable latent space of all the trained TNDM models.

The irrelevant factors in TNDM show task-aligned dynamics that do not depend strongly on the task type, but are rather homogeneous (see Figure 3). For instance, over the course of each trial, irrelevant factor 2 has a large initial fluctuation followed by a steady increase in its absolute value over time until around 600ms where it tapers off (around when the monkey reaches the target destination). As this factor is agnostic to the reach direction, this may reflect dynamics associated with execution of these movements more generally.

## 6 Discussion

In this work, we introduce TNDM, a nonlinear state-space model designed to disentangle the behaviourally relevant and behaviourally irrelevant dynamics underlying neural activity. We evaluated TNDM on synthetic data and on neural population recordings from PMd and M1 paired with a 2D center-out reach behaviour. We showed that TNDM was able to extract low-dimensional latent

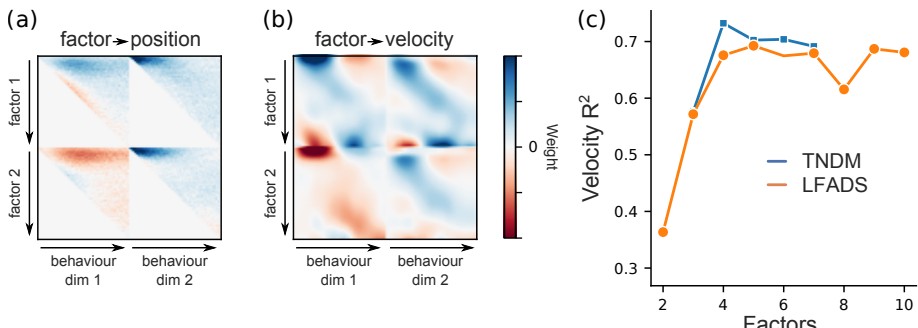

Figure 4: (a) Visualization of the weights of the TNDM behaviour decoder $C_y$ that transforms two relevant latent factors into behaviour (hand position). The behaviours are aligned horizontally and the factors vertically. The upper-triangular structure reflects causal decoding, i.e. factors can only influence future behaviour. The model had two relevant and two irrelevant factors. (b) Weights obtained by using ridge regression to predict movement velocity (x and y components) from the relevant factors. A diagonally banded structure can be observed indicating a (delayed) identity transformation. Unlike in (a), this matrix is not constrained to be causal. (c) Decoding accuracy for velocity obtained using ridge regression for LFADS and TNDM with two relevant factors and a varying number of irrelevant factors.

dynamics that were much more predictive of behaviour than those of current state-of-the-art state-space models without sacrificing its fit to the neural activity. This led us to the interpretation that the dimensionality of the neural activity associated with the 2D reaching task is potentially lower than previously thought and may be associated with the velocity of the hand.

Although the initial results presented for TNDM are quite promising, the method has a few limitations that should be addressed. First, we find that some hyperparameter settings combined with certain random initialisations can cause biologically implausible oscillations in the learned latent dynamics. While more work needs to be done to understand this, it could be related to the weighting between the behavioural and neural likelihoods or to the capacity of the model. A second limitation of TNDM is whether the disentanglement penalty between the relevant and irrelevant dynamics is sufficient. Although the covariance penalty works well in practice (on the presented datasets), disentangling sources of variation using deep generative models is still an open problem [16]. Similarly to PSID, TNDM could be implemented in multiple separate stages which may allow for better disentanglement of the relevant and irrelevant dynamics [28]. Third, the linear causal decoder we introduce for behaviour reconstruction is parameter inefficient: the number of parameters scales quadratically with time and dynamics/behaviour dimension. Lastly, it can be challenging to determine the 'correct' latent dimensionalities for TNDM. For the datasets in this paper, we found that performing a wide sweep over a number of relevant and irrelevant dimensionalities and then choosing the relevant dimensionality where the behaviour prediction saturates is a potential recipe for finding an interpretable latent space.

In future work, we plan to train TNDM with higher-dimensional behaviours such as joint angle or electromyography (EMG) recordings. We also plan to extend TNDM to non-temporal/discrete behaviours which are of interest in behavioural neuroscience (e.g. decision making). Finally, we hope to extend TNDM such that it can model dynamics with external inputs from another brain region, i.e. non-autonomous dynamics.

## 7 Broader Impact

In this work, we develop an expressive and interpretable model of neural population activity. As such, we imagine that TNDM will be useful for answering important questions about neural function (e.g. how the motor cortex gives rise to behaviour). We also imagine that the ability of TNDM to accurately model both the neural activity and the behaviour of interest will be of interest for the brain-computer interface community. We believe that TNDM (or the principles behind it) can be used to improve behaviour decoding from neural activity. We also hope that TNDM inspires more research into deep generative models of neural activity that incorporate in external variables of interest. A

possible negative societal impact of TNDM is that, like all deep neural network models, it requires a relatively large amount of compute and has a noticeable carbon footprint.

## Acknowledgements

We thank Alessandro Facchin and Nina Kudryashova for the code contributions and for the insightful discussions. We also thank the reviewers for their thoughtful critiques and suggestions.

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
