# Appendix

## 1 Behaviour decoder

As mentioned in Section 3, the choice of behaviour decoder can dramatically change the learned relevant latents. Figure 5 shows TNDM trained with both the one-to-one Gaussian behaviour decoder introduced in PSID and with the flexible, casual decoder introduced in this work. As can be seen with the one-to-one Gaussian decoder, TNDM's behaviourally relevant factors perfectly replicate the behaviour. This is because there is so little flexibility for the decoder to model the behaviour that replicating the behaviour is the best option (despite the potential negative effect to neural reconstruction). This forces the irrelevant factors to encode behavioural information as the neural reconstruction will largely depend on the irrelevant factors (see Figure 6). With the causal linear decoder, TNDM is able to extract behaviourally relevant factors that explain the behaviour while still contributing meaningfully to the neural reconstruction (see Figure 6). The irrelevant factors, in this case, do not need to encode behavioural information and can instead encode aspects of the neural activity unrelated to the specific behaviour. We find that with only one irrelevant factor, the neural reconstruction suffers for the one-to-one decoder and improves steadily as you add more irrelevant factors.

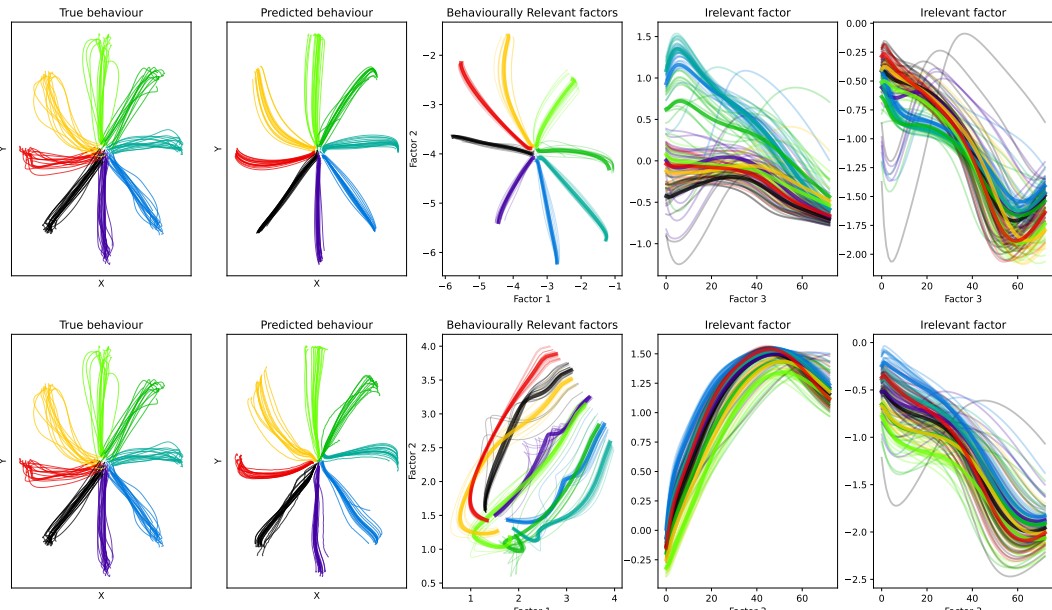

Figure 5: Visualizations of the predicted behaviour and latent factors for TNDM trained with two different decoders. On top, TNDM is trained with a one-to-one Gaussian decoder (introduced in Section 2). As can be seen, the lack of flexibility in the decoder forces the relevant factors to simply replicate the behaviour. This means that the irrelevant factors have to encode behavioural information since they are primarily used for neural reconstruction. On bottom is the causal linear decoder introduced in this work. Here, the relevant factors capture more structure in the neural activity while still allowing for good behaviour reconstruction. This lets the irrelevant factors encode less behavioural information.

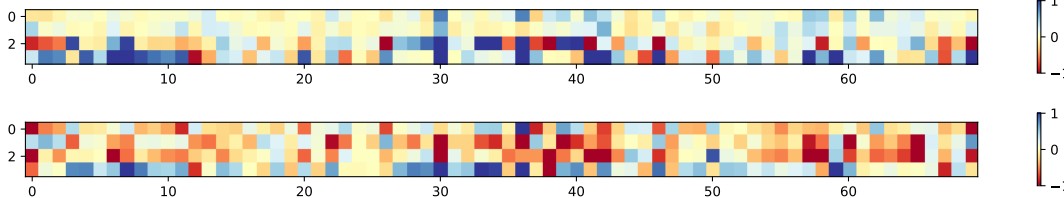

Figure 6: Visualization of the learned neural reconstruction weight matrices for the factors shown in Figure 5. These matrices transform the learned factors into neural firing rates. On top is the neural weight matrix for TNDM trained with a one-to-one Gaussian decoder and on bottom is the neural weight matrix for TNDM trained with the linear causal decoder. In both cases, their are four factors that are transformed into the firing rates of 70 neurons (the top two factors are relevant and the bottom two are irrelevant). Interestingly, for TNDM trained with the one-to-one decoder, the relevant factors are barely used for neural reconstruction (i.e. low magnitude weights). This implies that the learned factors are not informative of neural activity and that the irrelevant factors are mainly being used. For TNDM trained with the linear causal decoder, however, the relevant factors play a much larger role in the neural reconstruction (i.e. higher magnitude weights).

## 2 Premotor cortex results

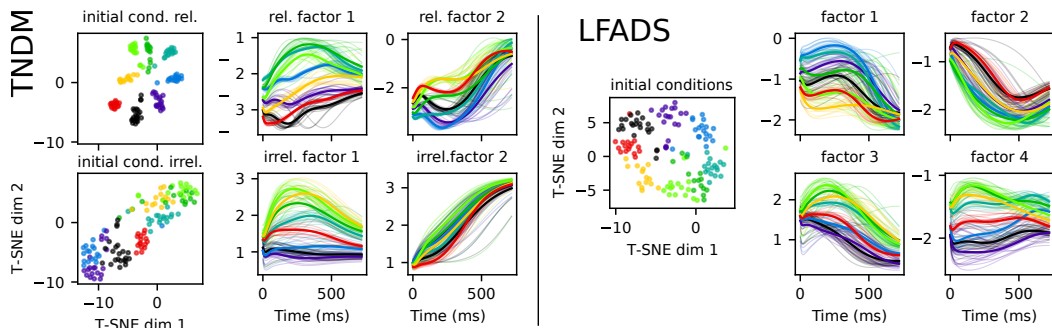

Figure 7: Visualization of initial conditions and latent dynamical factors for a model of premotor cortex (PMd) activity. The activity was recorded simultaneously with the motor cortex (M1) activity shown in Figure 3, main text. As in that figure, TNDM (left) has two relevant and two irrelevant factors, and LFADS (right) four factors. On the left inferred initial conditions for each method are shown after reducing the dimension to 2 with T-SNE. There is a clear distinction between the conditions relating to different movement directions (indicated by different colours) in the relevant factors, but unlike for M1 the irrelevant factors also contain some structure that distinguish trial types. The LFADS initial conditions for PMd show some distinction between behaviours, but similar to the M1 data this is much weaker than for the relevant factors in TNDM. Condition-averaged inferred latent dynamical factors (along with the single-trial factors; plots on the right) again show a clear distinction between different behaviours for the relevant factors, while this information is mixed in the irrelevant factors. As for M1, there is no clear behaviour separation in the factors of LFADS.

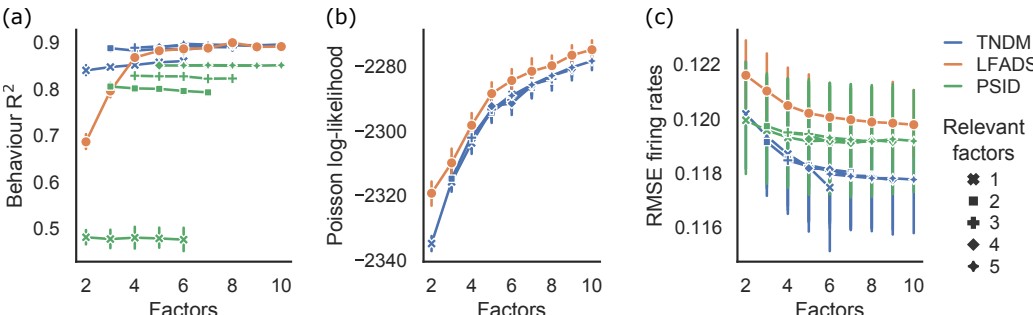

Figure 8: Behaviour and activity reconstruction accuracy for TNDM, LFADS, and PSID for monkey premotor cortex (PMd) activity during the center-out reaching task. The data used here was recorded simultaneously with the M1 activity shown in Figure 2 (main text). Each plot shows performance as a function of the total number of latent factors, averaged over five fits with different initialisation (random seeds) and different random training-test data splits. Error bars show standard error of the mean. For TNDM and PSID, the reconstruction performed by the behaviour decoder (with only the relevant factors) is shown, while for LFADS a ridge regression was used to decode the behaviour ex post factor. Behaviour reconstruction and log-likelihood were computed on held out test data, and the firing rate RMSE on the whole data set to allow for more reliable averaging. Similar to M1, TNDM requires at least two relevant factors for saturating behaviour reconstruction accuracy for all model sizes. LFADS gradually reaches peak accuracy, saturating at 5 factors, and PSID requires at least five relevant factors. As in M1, neural activity reconstruction in TNDM solely depends on the total number of factors, irrespective of the fraction of relevant factors.

# 3 Hyperparameters

Table 1 shows the hyperparameters of LFADS and TNDM used for the main experiments. We did not run an exhaustive search over these parameters. For LFADS, we used default parameters for all regularization terms. For TNDM, we used a small value for $\lambda_b$ such that the behaviour likelihood was smaller than the neural likelihood (the neural reconstruction was the primary goal). The dropout for TNDM was set to be slightly higher than for LFADS so as to not overfit the behaviour; we found that this higher dropout was not helpful for LFADS. We also set the batchsize to 16 for TNDM and 10 for LFADS; we found that LFADS latent factors were less informative about behaviour when using a higher batch size. We qualitatively found that both models provided a good fit to the neural data with these parameters (see Figures 3, 8)

Table 1: Hyperparameters of LFADS and TNDM (adapted from [17]).
'N' - number of units in generator (irrelevant generator for TNDM). 'rel N' - number of units in relevant generator. 'g0' - initial conditions (irrelevant initial conditions for TNDM). 'rel g0' - relevant initial conditions. 'E' - encoder. 'G' - decoder (irrelevant decoder for TNDM). 'rel G' - relevant decoder. '$\lambda_b$' - weight for behaviour likelihood. '$\lambda_Q$' - weight for disentanglement loss. 'KP' - keep probability for dropout. 'B' - batch size.

| Model | N | rel N | g0 E dim | rel g0 E dim | G L2 | rel G L2 | KP | $\lambda_b$ | $\lambda_Q$ | B |
|-------|-----|-------|----------|--------------|------|----------|-----|-------|-------|-----|
| LFADS | 64 | N/A | 64 | 64 | 2000 | N/A | .95 | N/A | N/A | 10 |
| TNDM | 64 | 64 | 64 | 64 | 2000 | 2000 | .85 | .2 | 1000 | 16 |

# 4  Ablation study of disentanglement penalty

The primary aim of TNDM is to disentangle the behaviourally relevant and the behaviourally irrelevant latent dynamics underlying neural activity. Although there will naturally be some separation between the two sets of factors in TNDM (since the behaviourally relevant factors must reconstruct the behaviour), the factors may still share information. To further encourage disentanglement of the relevant and irrelevant factors, we introduce a disentanglement penalty on the initial condition distributions (described in Section 3 of the main text). Although we did not confirm the efficacy of this penalty exhaustively, in Figure 9 we show TNDM with and without the disentanglement penalty for a specific example. As can be seen, with 2 relevant and 2 irrelevant factors, the penalty forces the irrelevant factors to encode less information about the behaviour and the relevant factors to encode less behaviourally irrelevant information. This is quantified in Table 2. Although there is some evidence that this disentanglement penalty is useful, there is room for improvement as it is only applied to the initial condition distributions and not the factors themselves.

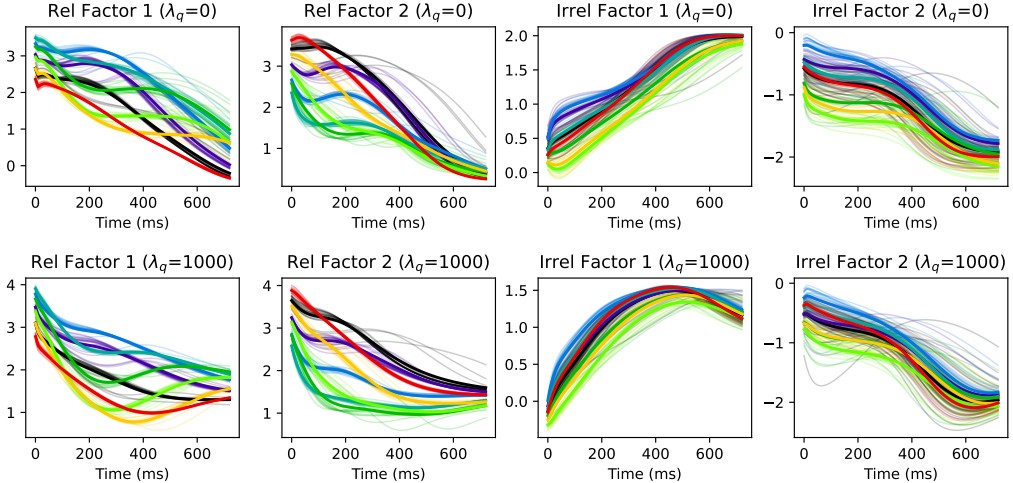

Figure 9: Visualizations of the relevant and irrelevant latents for two runs of TNDM with and without the disentanglement penalty. In each box, we plot the condition-averaged inferred latent dynamical factors (along with the single-trial factors). On top, TNDM is applied to the neural data and behaviour with no disentanglement penalty. On bottom, TNDM is applied to the neural data and behaviour with a disentanglement penalty that is weighted by $\lambda_Q = 1000$. As can be seen, when the disentanglement penalty is applied, the irrelevant factors contain less behavioural information (condition-averaged irrelevant latents are less separated). Also, the relevant factors seem to share less information with the irrelevant factors.

Table 2: Behavioural prediction using the relevant and irrelevant factors shown in Figure 9. To determine how much behavioural information is stored in the relevant and irrelevant factors, we regress the inferred training relevant and irrelevant latents (with a linear ridge regression) to the training behaviours . We then report the test $R^2$ of the regression using the test relevant and irrelevant latents and the test behaviours. As can be seen, the irrelevant factors learned by TNDM with no disentanglement penalty contain more behavioural information (higher test $R^2$).

| Rel Facs ($\lambda_Q = 1000$) | Irrel Facs ($\lambda_Q = 1000$) | Rel Facs ($\lambda_Q = 0$) | Irrel Facs ($\lambda_Q = 0$) |
|---|---|---|---|
| 0.890 | 0.245 | .883 | 0.373 |

# 5 Preferential subspace identification failure mode

In this supplement, we perform two experiments that provide evidence that PSID does not learn latent dynamics that are informative about the neural activity for our M1 dataset. As there are very few adjustable hyperparameters for PSID, we only had to set the number of block-rows in (i.e. "future horizon" and "past horizon") and the smoothing method for the spiking data. For the horizon, we set the value to the default of 10 (although changing this parameter seemed to have little effect) and for the smoothing we used a Gaussian filter with a standard deviation of 50 ms. We perform our analysis of PSID with 2 behaviourally relevant factors. All code for these analyses can be found at https://github.com/HennigLab/psid_technical_report.

The first experiment we run is training PSID normally on the center-out reach dataset and then inspecting the learned state-transition matrix A. According to the Sani et al. [25], the state-transition matrix $A$ "[characterizes] the behaviourally relevant neural dynamics". As can be seen in Table 3 and Figure 10, the learned state-transition matrix $A$ is approximately the identity with eigenvalues that have real value 1 and an insignificant complex component. To better understand if the state-transition matrix $A$ being an identity matrix still meaningfully characterizes the neural activity, we also train PSID on time shuffled and trial shuffled neural data. In both cases, the state-transition matrix $A$ is again approximately the identity matrix with an insignificant complex component. These experiments suggest that the learned identity matrix is not informative about the neural activity for PSID.

We postulate that the state-transition matrix $A$ is uninformative about the neural activity due to the nonlinear behaviour. The behaviour is nonlinear across all trials due to the 8 different reach directions. To test if this is the case, we train PSID multiple times with 1 to 8 reach directions. As can be seen in Table 4 and Figure 11, the state-transition matrix $A$ matrix quickly collapses to the identity matrix as the number of reach directions increases past 1. This implies that the multi-reaching behaviour is difficult for PSID to model with linear dynamics.

Table 3: In this table, we show the eigenvalues of the state-transition matrix A for PSID trained with 2 behaviourally relevant factor and 0 behaviourally irrelevant factors (the lack of behaviourally irrelevant factors should have no affect on the behaviourally relevant factors). When trained normally, with trial shuffled data, and with time shuffled data, the eigenvalues of A is always close to 1 with an insignificant complex component.

| Experiment | Normal | Trial Shuffled | Time Shuffled |
|---|---|---|---|
| Eigenvalues of A | 1.01 + 0.0016j, 1.01 - 0.0016j | 1.01 + 0.0015j 1.01 - 0.0015j | 1.016, 1.018 |

Table 4: In this table, we demonstrate how the state-transition matrix A of PSID approaches the identity matrix as the number of reach directions increases. We again ran this experiment for PSID trained with 2 behaviourally relevant factors and 0 behaviourally irrelevant factors.

| # reach directions | 1 | 2 | 3 | 4 | 5 | 6 | 7 | 8 |
|---|---|---|---|---|---|---|---|---|
| 2 norm of A matrix | 1.06 | 1.01 | 1.01 | 1.01 | 1.01 | 1.01 | 1.01 | 1.01 |

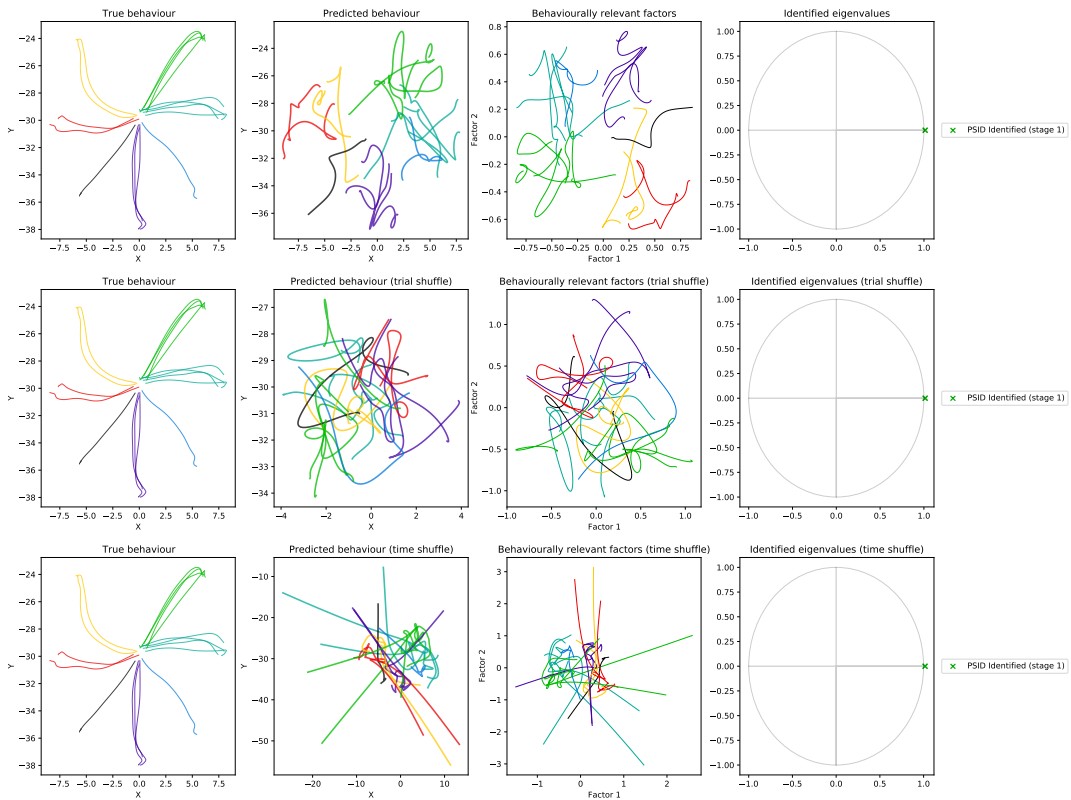

Figure 10: Visualizations of the relevant latents and behavioural predictions of PSID (using a Kalman filter prediction approach) on regular and shuffled neural data. On top, visualizations are shown from PSID when trained on the neural data and behaviour normally. The behaviour prediction resembles random walks and the $A$ matrix is close to the identity. In the middle and the bottom plots, visualizations are shown from PSID when trained on shuffled neural data (by trial and in time) and behaviour. While the behaviour prediction is significantly worse, the $A$ matrix is again close to the identity. These experiments suggest that PSID is finding latent dynamics that are uninformative about the neural activity.

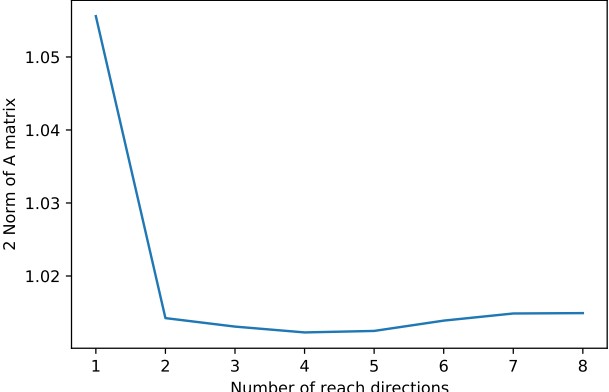

Figure 11: Visualization of the 2 norm of the state-transition matrix $A$ when PSID is trained on 1 to 8 different reach directions. As the number of reach directions increases past 1, the 2 norm quickly collapses to 1 which implies that the A matrix is close to the identity. This experiment suggests that the nonlinear multi-reaching behaviour is challenging for PSID to model as it is a linear dynamical model.

# 6 Simulated data results

To validate TNDM on simulated data, we run both TNDM and LFADS on synthetic spike trains generated from a Lorenz system, a common benchmark for state space models of neural activity. The spike trains are stochastically generated from a 3-dimensional Lorenz system which is partitioned into behaviourally relevant and irrelevant factors. We transform the relevant factors into the behaviours using a linear transformation and then we transform all the factors into the neural firing rates using a separate linear transformation. The neural firing rates are then transformed into a spike train through an exponential nonlinearity and a Poisson random variable. For our analysis, we set the number of neurons to 30, the number of behaviour dimensions to 4, and the number of behaviourally relevant factors to 2 (out of 3). The initial conditions for the Lorenz system are sampled from a Uniform distribution and the behaviour is corrupted with additive noise sampled from a standard Normal distribution. The code we used for generating the synthetic spike trains can be found at: https://github.com/HennigLab/tndm/blob/main/tndm/lorenz/lorenz_generator.py.

For training, we evaluate the performance of TNDM and LFADS when trained with 50, 100, and 200 trials. Each trial consists of a single initial condition that is evolved into the three latent factors and then into the behaviour and neural activity. We also evaluate each model across three baseline neural firing rates: 5, 10, and 15 Hz. The results are summarized in Table 5. TNDM is competitive with LFADS across all numbers of trials and baseline firing rates. There is evidence that TNDM outperforms LFADS on the lowest firing rate trials, however, we did not perform an exhaustive hyperparameter search so it is likely that the hyperparameters of LFADS can be adjusted to obtain better results in these cases.

For this analysis, as the relationship between the latent factors and the behaviour has no time lag, we utilize a behaviour decoder for TNDM that has no time lag. This is in contrast to the decoder that we use on real data which allows for capturing arbitrary lags. Also, we utilize a Tensorflow2 re-implementation of the original TNDM and LFADs model for this analysis. These re-implementations can be found at the following repository: https://github.com/HennigLab/tndm. We are currently working on improving and extending this implementation so the commit that should reproduce this analysis is 58a0a71b529f5fbe72ce3f6516daed83ce5885ca.

Table 5: In this table, we report the results of TNDM and LFADS when run on the synthetic Lorenz system data. The average $R^2$ of TNDM and LFADS is reported for 3 runs of each model on each of the training conditions.

| Firing Rate | 5 | | | 10 | | | 15 | | |
|---|---|---|---|---|---|---|---|---|---|
| Train Trials | 50 | 100 | 200 | 50 | 100 | 200 | 50 | 100 | 200 |
| LFADS | .52 | .86 | .88 | .54 | .88 | .92 | .50 | .87 | .92 |
| TNDM | .71 | .83 | .88 | .66 | .86 | .92 | .67 | .86 | .93 |

# 7 Leave one direction out results

To illustrate that TNDM learns an interpretable and meaningful latent space, we run an experiment where we leave out one reach direction during training. We train TNDM on 7 out of the 8 reach directions and then we see if we can infer the initial conditions, latent factors, and behaviour of the held-out reach condition. The results are shown in Figure 12. Although not perfect, TNDM recovers initial conditions, latent factors, and behaviours for the held-out reach condition that are close to the model that is trained with all 8 reach conditions. This suggests that TNDM is able to learn latent dynamics that meaningfully capture the behavioural/neural manifold of reach. We imagine this result can be improved with more trials ($\sim$100 trials is quite limited) and with more hyperparameter tuning.

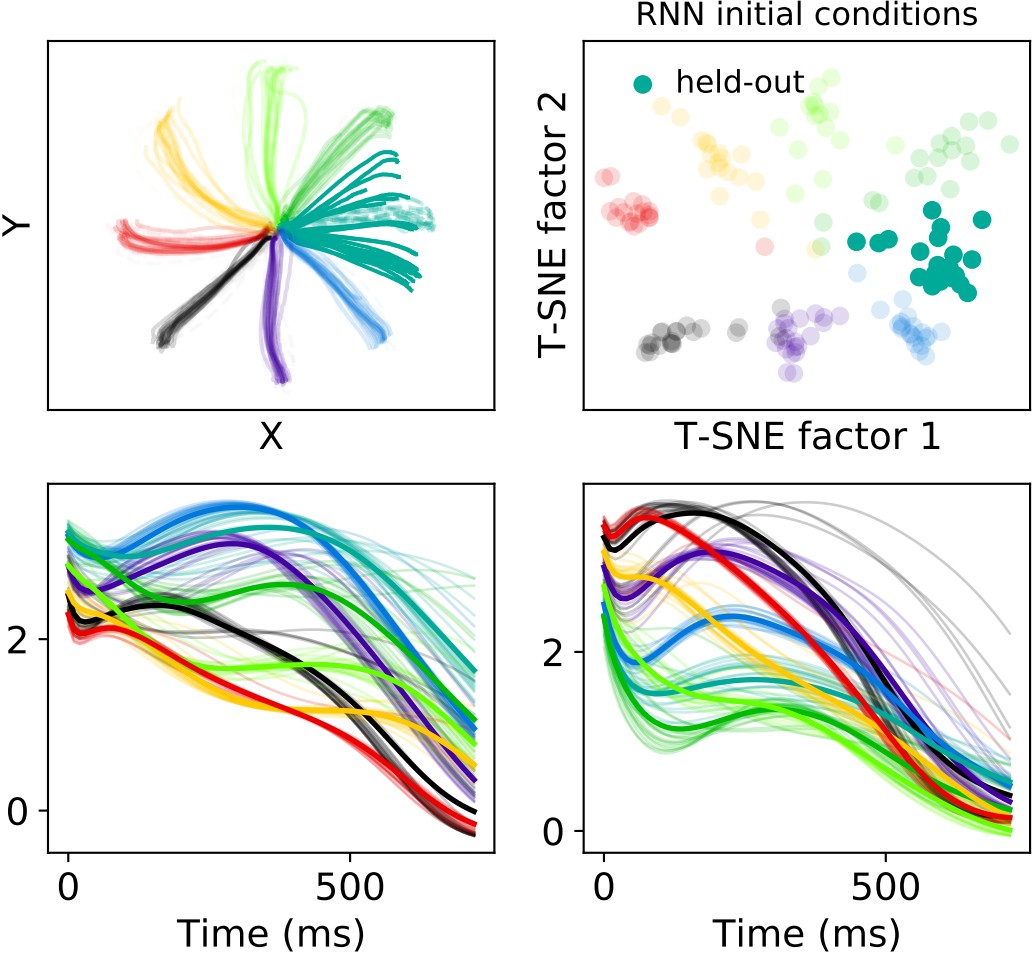

Figure 12: We train TNDM on 7 out of the 8 reach directions and then we see if we can infer the initial conditions, latent factors, and behaviour of the held-out reach condition. TNDM is able to recover these fairly well despite the small number of training trials ($\sim$100 trials).