# OpenReview forum: "Targeted Neural Dynamical Modeling"
_NeurIPS.cc/2021/Conference — NeurIPS 2021 Poster_

### Official Review · Reviewer_1AfS · 2021-07-11

**Rating:** 5
**Confidence:** 4

**Summary:**

The authors introduce a new sequential VAE that simultaneously models behavioral dynamics and neural activity for experimental neuroscience data. The model includes partitioned latents which map to either neural activity and behavior, or just behavior.  The model uses RNNs to describe the evolution of each of these latent states (similar to LFADS), and the model is trained by optimizing an objective function which contains separate variational distributions for behaviorally relevant and irrelevant latents, as well as a penalty to prevent entanglement of the latent states. The authors test their model on neural data recorded from Monkey V1 during a reaching task.

**Main Review:**

I think this work is an interesting application and extension of sequential VAEs for neuroscience. The writing is clear, the model is novel, and I believe the work would be of interest to the neuroscience community. My primary concern is that this paper, in its current form, may not be appropriate for NeurIPS.  My reason for this is that there is not a lot of mathematical or theoretical development -- specifically, the objective function is weakly motivated, and there is minimal to no attention given to inference. I elaborate on each of these pieces below.

Weak motivation of an objective function: The introduction of the ELBO in (3) feels like it naturally follows from the previous explanation and description of the model.  However, immediately after introducing this, the authors say that the variational distributions can become entangled and thus introduce the cross-correlation penalty (4). Though they show some qualitative comparisons with and without this penalty in the appendix, the motivation feels ad hoc and quickly thrown in. As this is a fundamentally new objective, I think elaboration is warranted.

I'm curious if the authors could bake this type of regularization into the model itself, as opposed to just modifying the objective function and exploring the qualitative changes. Perhaps one could impose a prior on the latent state(s)?

Additionally, the authors introduce a parameter balancing the behavioral and neural objectives (4) with minimal justification or motivation.

Lack of attention given to inference: The current inference section is mostly devoted to specifying the objective function, not actually how inference is done. I am not familiar with inference procedure in sequential VAEs. Does it use the reparameterization trick? Is there anything specific in this model that can be done to perform inference quickly or efficiently?

I do think that overall this work is interesting and well-done.  However, given that the work is a somewhat natural combination pieces of existing models, additional attention should be devoted to expanding on partitioned seq VAEs in general, some more fleshing out of inference, or better theoretical justification/exploration of the objective function, especially for NeurIPS.

A few other smaller concerns:

The authors fail to mention statespace models that more have flexible than linear dynamics - namely GPFA. See

Zhao, Yuan, and Il Memming Park. "Variational latent gaussian process for recovering single-trial dynamics from population spike trains." Neural computation 29.5 (2017): 1293-1316.

Keeley, Stephen, et al. "Efficient non-conjugate Gaussian process factor models for spike count data using polynomial approximations." International Conference on Machine Learning. PMLR, 2020

The authors talk about the importance of time-lags in the model, but there are no results supporting this.

**Time Spent Reviewing:**

4 hours

---

> ### Author Response · Authors · 2021-08-10
> **Response to Reviewer 1AfS**
>
> ### General
>
> We thank reviewer 1AfS for their thorough and thoughtful review. We appreciate that the reviewer recognized the novelty of our model and believed our work would be of interest to the neuroscience community. It appears that the reviewer’s main concerns are in the motivation of the objective function (both the disentanglement penalty and parameter for balancing the behavioral/neural objectives) and in the lack of attention given to the inference in our manuscript. Hopefully, we can address both of these concerns satisfactorily in the following response and in our updated manuscript if accepted.
>
> Before responding to specific concerns, however, we believe it is important to note that the major contribution of TNDM is to propose a novel probabilistic graphical model for learning the non-linear behaviourally relevant and irrelevant dynamics that underlie behaviour and neural activity. While the exact form of the disentanglement penalty and objective may change with future iterations of the work (unsupervised disentangled representation learning is still an unsolved problem (Locatello 2018)), we believe the proposed model is still worth exploring and developing further within the greater neuroscience community.
>
> Citation
> * Locatello, Francesco, et al. "Challenging common assumptions in the unsupervised learning of disentangled representations." international conference on machine learning. PMLR, 2019.
>
> ### Major Concerns
>
> > My primary concern is that this paper, in its current form, may not be appropriate for NeurIPS. My reason for this is that there is not a lot of mathematical or theoretical development...
>
> As mentioned above, the major contribution of TNDM is the proposed probabilistic graphical model that allows for learning non-linear behaviourally relevant and irrelevant dynamics from behaviour and neural activity. We believe that TNDM’s novel probabilistic graphical model, architecture, and application to paired neural activity and behaviour would be an appropriate fit for NeurIPS as it greatly improves upon the state-of-the-art in latent variable modelling for neural population activity. As mentioned in the general response, we also believe that TNDM’s prediction about M1 activity during reach, that the behaviourally relevant dynamics represent information about movement velocity, is an important contribution and of interest for the neuroscience community/NeurIPS.
>
> > Weak motivation of an objective function: The introduction of the ELBO in (3) feels like it naturally follows from the previous explanation and description of the model. However, immediately after introducing this, the authors say that the variational distributions can become entangled and thus introduce the cross-correlation penalty (4). Though they show some qualitative comparisons with and without this penalty in the appendix, the motivation feels ad hoc and quickly thrown in.
>
> While this is a valid criticism, we believe that it has less to do with our model and more to do with our writing: *we did not provide the proper context/background about disentangled representation learning to make our motivation clear*. In unsupervised disentangled representation learning using VAES, it is standard to first introduce the ELBO for the model and then introduce additional penalties that encourage disentanglement/independence of the latent representations (Kumar et al. 2017, Chen et al. 2018, Kim and Mnih 2018). Importantly, these additional penalties should be applied in such a way that the final objective is still a valid lower bound of the log likelihood which our cross-correlation penalty (which is always negative) ensures. The cross-correlation penalty we utilize is a simple way of reducing information sharing between the two distributions by penalizing the linear correlation of the two distributions. As this penalty is also relatively standard, we decided to only provide a qualitative ablation study in Supplement 5. We would be happy to add more details about disentangled representation learning and the cross-correlation penalty into the manuscript to better motivate this objective.
>
> Citations:
> * Kumar, Abhishek, Prasanna Sattigeri, and Avinash Balakrishnan. "Variational inference of disentangled latent concepts from unlabeled observations." arXiv preprint arXiv:1711.00848 (2017).
> * Chen, Ricky TQ, et al. "Isolating sources of disentanglement in variational autoencoders." arXiv preprint arXiv:1802.04942 (2018).
> * Kim, Hyunjik, and Andriy Mnih. "Disentangling by factorising." International Conference on Machine Learning. PMLR, 2018
>
> > I'm curious if the authors could bake this type of regularization into the model itself, as opposed to just modifying the objective function and exploring the qualitative changes. Perhaps one could impose a prior on the latent state(s)?
>
> Our choice of architecture and model was actually very important for encouraging the disentanglement of the two sets of dynamics. Originally, TNDM consisted of a single latent distribution for the initial conditions and two RNN decoders that were used to transform those initial conditions into the behaviourally relevant and irrelevant dynamics. With this model structure and shared latent space, we found that the irrelevant latent factors always contained large amounts of information about the behaviour despite having two separate RNN decoders. By having separate latent spaces for the initial conditions and by penalizing information sharing between these latent spaces, we were better able to isolate the two sources of dynamics. Also, to be clear, many modifications to the objective function in disentangled representation learning can be interpreted as imposing priors on the latent states like, for example, imposing a factorial prior on the latent states (Chen et al. 2018).
>
> Citations:
> * Chen, Ricky TQ, et al. "Isolating sources of disentanglement in variational autoencoders."
>
> > The authors introduce a parameter balancing the behavioral and neural objectives (4) with minimal justification or motivation.
>
> It is true that we do not spend much time justifying our choice to introduce a hyperparameter for balancing the reconstruction terms. However, to the best of our knowledge, having a hyperparameter for balancing multiple likelihood terms is standard when dealing with heterogeneous data types and multiple likelihood terms. As discussed in Ma et al. 2020, applying VAEs to mixed type heterogenous data is not straightforward because the contribution of each likelihood term will be very different which gives rise to a challenging optimization problem. A potential approach to dealing with this imbalance is to introduce hyperparameters for scaling the likelihood terms appropriately (Ma et al. 2020, Whiteway et al. 2021). We would be happy to update our manuscript with these sources to demonstrate why we chose to introduce a simple scaling term for the likelihoods. Also, we can include an additional supplement analyzing the effect of varying this hyperparameter.
>
> Citations:
> * Ma, Chao, et al. "VAEM: a deep generative model for heterogeneous mixed type data." arXiv preprint arXiv:2006.11941 (2020).
> * Whiteway, Matthew R., et al. "Partitioning variability in animal behavioral videos using semi-supervised variational autoencoders." bioRxiv (2021).
>
> > Lack of attention given to inference: The current inference section is mostly devoted to specifying the objective function, not actually how inference is done
>
> We agree that the lack of attention in our manuscript given to the inference is problematic as sequential VAEs are not commonly used or discussed in literature. To clarify to the reviewer, the only stochastic variables of TNDM are the initial conditions of the behaviourally relevant and irrelevant dynamics. Inferring the initial conditions is done with a feed-forward pass through our RNN encoder followed by two separate linear transformations to recover the variational posterior parameters of the behaviourally relevant and irrelevant initial conditions (i.e. means and variances). The reparameterization trick is used to sample from each initial condition distribution and then the sampled initial conditions are evolved using separate decoder RNNs to produce the behaviorally relevant and irrelevant high-dimensional dynamics. The high-dimensional dynamics at each time-step are projected into a low-dimensional subspace to recover the low-dimensional ‘dynamical’ factors which we analyze throughout the paper. We realize now that our plate diagram in Figure 1 is wrong and potentially quite confusing as it implies that the dynamical factors are stochastic despite the fact that they are a deterministic transform of the initial conditions which are actually stochastic. Hopefully this clarifies to the reviewer how inference is performed with our model. We would be happy to add a few additional sentences about the inference of our model into the Inference section and also to fix our plate diagram so that it more accurately reflects the true structure of our model in the camera ready paper.

---

> > ### Author Response · Authors · 2021-08-10
> > **Response to Reviewer 1AfS continued.**
> >
> > ### Minor concerns
> >
> > > The authors fail to mention state-space models that more have flexible than linear dynamics - namely GPFA.
> >
> > We purposefully did not discuss GPFA or any other Gaussian Process-based latent variable models in our manuscript because these methods are not state-space models. These methods are not state-space models because they do not explicitly learn the transition function of an underlying dynamical system, but rather model the statistics of the latent processes directly (Rutten et al. 2020, Hurwitz et al. 2021). We would be happy to add an additional sentence to the manuscript about our choice to focus on state-space models exclusively.
> >
> > Citations:
> > * Rutten, Virginia, et al. "Non-reversible Gaussian processes for identifying latent dynamical structure in neural data." Advances in Neural Information Processing Systems (2020).
> > * Hurwitz, Cole, et al. "Building population models for large-scale neural recordings: opportunities and pitfalls." arXiv preprint arXiv:2102.01807 (2021).
> >
> > > The authors talk about the importance of time-lags in the model, but there are no results supporting this.
> >
> > In Supplement 1, we do provide some evidence of the importance of capturing time-lags. In this experiment, we compare the no time-lag decoder to the full causal decoder. We show that the behaviourally relevant factors learned by TNDM with the full causal decoder contribute more meaningfully to the neural reconstruction than when using the no time-lag decoder. Generally we expect a same-time behaviour decoding to fail due to inherent (and variable) neuronal and kinetic latencies between activity and observable behaviour. We agree, though, that the paper would benefit from additional results supporting our choice of decoder.

---

### Official Review · Reviewer_wP6X · 2021-07-14

**Rating:** 6
**Confidence:** 5

**Summary:**

In this work, the authors propose a state-space model and inference algorithm that aims to recover the latent nonlinear dynamics from sequential data with particular interests in neuroscience. The proposed model inherits the latent factor analysis via dynamical systems (LFADS) and incorporates the advantage of disentangling behavior relevant/irrelevant dynamics in the latent space as preferential subspace identification (PSID) does. It overcomes the limitation of linear dynamics and more faithfully captures the behavior. The proposed method was tested on the data from a monkey reaching task.

**Limitations And Societal Impact:**

The authors discussed the limitations and broader impact.

**Main Review:**

Originality: The proposed model uses the same structure as LFADS with extra conditional Gaussian observation model for behavior variables of the similar idea as that of PSID. The objective function modifies the ELBO with terms that regularize the entanglement and behavior likelihood. Previous work are discussed and compared. The contributions are clearly stated.

Quality: There're more have to be done to support the claims.
- Only one empirical result on the reaching task was shown. It isn't adequate to evaluate the strength and weakness of the method. Synthetic data and other tasks are needed.
- The work is focusing on the reaching task overly. TNDM seems to be tightly tailored for the task. These particular settings prohibits the method from generalizing to other task and weaken its easiness for use. E.g. the choice of initial conditions is needed. It is also mentioned that a random initialization might lead to weird dynamics; The causal relationship is not always true for behavior and neural activity, considering feedback scenarios.
- "to learn low-dimensional dynamics" is ambiguous. In TNDM, the dynamical system is modeled by a high-D RNN as well as LFADS. Only the latent factor $z$ are low-D.
- The power of SSM like LFADS is not only to recover the latent trajectories but also to learn the dynamical system (velocity/phase portrait/simulated trajectories). This should be also evaluated.
- It is expected that TNDM outperforms LFADS on behavior encode at small number factors, because of imperfect encoding of behavior variables and other internal dynamics.
- It is also important to evaluate the disentanglement, e.g. the decoding performance of irrelevant dynamics.
- PSID used Gaussian filtered spikes. This made the data and likelihood are both different from that of TNDM. Since TNDM is not limited to Poisson for spikes. It's more fair to fit TNDM to the same data and likelihood as PSID.

Clarity: The submission is well organized. The writing can be improved.
- The authors spent a lot for the details of LFADS and PSID but not say much about how PSID separates dynamics which is a key point to the work.
- In Eq(2), the encoder network is only a function of the spike $x$ without behavior $y$? and why not?
- It is unclear that how the initial conditions $\Phi_r$ and $\Phi_i$ are parametrized in Eq(2).
- Where are $\mathbf{g}_{0r}$ and $\mathbf{g}_{0i}$ defined? These bold symbols have different indexing order from their plain counterparts. Are they vectors?
- How is the mean of the sample cross-correlations between $\mathbf{g}_{0r}$ and $\mathbf{g}_{0i}$ computed?

Significance: The results show  that TNDM can capture, from neural recordings, the latent trajectories that faithfully reconstructs the neural activity and behavior. It could help to separate the latent factors into behavior-related and internal dynamics. The demonstration could be improved though.

Minor comments:
- Why to trim the trials to minimum? Varying length trials should not be a prolem.
- Why the covariance of behavior is fixed at the identity matrix?

**Time Spent Reviewing:**

10

---

> ### Author Response · Authors · 2021-08-10
> **Response to Reviewer wP6X**
>
> ### General
>
> We thank reviewer wP6X for the highly detailed and comprehensive review. We appreciate the amount of time they spent giving feedback on our paper and we hope we can adequately address the concerns listed by the reviewer about the quality and clarity of our paper in the following response. At the suggestion of the reviewer, we have run additional experiments with synthetic data to validate TNDM empirically.
>
> ### Major Concerns
>
> > Only one empirical result on the reaching task was shown. It isn't adequate to evaluate the strength and weakness of the method. Synthetic data and other tasks are needed.
>
> In our manuscript, we actually provide empirical results for two datasets: (1) paired M1 activity and 2D hand position and, (2) paired PMd activity and 2D hand position. We agree, however, that this may not be enough to validate our method. To this end, we have also evaluated TNDM on synthetic spike trains generated from a Lorenz system, a common benchmark for state space models of neural activity. To generate the neural activity and behaviour, we partition the latent factors into behaviorally relevant and irrelevant factors and then perform a random transformation of the appropriate factors. We then asked how well these latent dynamics are reconstructed by each model. The average $R^2$ (for 3 runs of each model with 1000 test trials) are as follows:
>
> | Firing rate |  5  |     |     |  10 |     |     |  20 |     |     |
> |:-----------:|:---:|:---:|:---:|:---:|:---:|:---:|:---:|:---:|:---:|
> | Train Trials      | 50  | 100 | 200 | 50  | 100 | 200 | 50  | 100 | 200 |
> | ---         | --- | --- | --- | --- | --- | --- | --- | --- | --- |
> | LFADS       | .52 | .86 | .88 | .54 | .88 | .92 | .50 | .87 | .92 |
> | TNDM        | .71 | .83 | .88 | .66 | .86 | .92 | .67 | .86 | .93 |
>
> This shows that TNDM matches the performance of LFADS when the number of trials available for training is sufficiently high. TNDM performs better when the number of trials is smaller, but we note that LFADS hyperparameters likely can be adjusted to obtain better results in these cases as well. Note that the behaviour and neural reconstruction are high for TNDM across all numbers of trials. We are happy to include this analysis in the supplement to show that TNDM is competitive in a standard benchmark.
>
>
> > The work is focusing on the reaching task overly. TNDM seems to be tightly tailored for the task. These particular settings prohibits the method from generalizing to other tasks and weaken its easiness for use. E.g. the choice of initial conditions is needed.
>
> We do not agree that TNDM is tightly tailored for reaching tasks and that it will not generalize to other tasks in experimental neuroscience. TNDM, LFADS and PSID all assume that neural population activity can be modeled as an autonomous dynamical system. TNDM and PSID also both assume that behaviour is generated by the behaviourally relevant latent dynamics. The main difference between TNDM and PSID is in the underlying form of the learned dynamics (non-linear vs. linear) and in TNDM’s assumption that the behaviourally relevant dynamics can give rise to the behaviour with arbitrary time-lags. Therefore, TNDM would be perfectly suited for any of the datasets provided, for instance, in Sani et al. 2021, including saccadic eye movements paired with visual PFC neural activity. Equally, TNDM can be used to analyze neural representations in navigation and choice tasks where the behavioural variable might be velocity, position, or even discrete variables such as licking frequency. We also have some results for TNDM on synthetic data which do not correspond to a reaching behaviour. We agree that not all brain areas can be modelled using autonomous dynamical systems and, much like LFADS, we would be interested in extending TNDM to model inputs from other brain areas.
>
> Citations:
> * Sani, Omid G., et al. "Modeling behaviorally relevant neural dynamics enabled by preferential subspace identification." Nature Neuroscience 24.1 (2021): 140-149.
>
> > It is also mentioned that a random initialization might lead to weird dynamics
>
> We do find that with a small number of latent variables, some random initializations lead to biologically implausible oscillations in the learned dynamics. This can be compensated for by enforcing smooth dynamics through regularization. However, we feel this is a standard issue when working with deep neural networks and deep generative models as they can be sensitive to random initialization. Through our experience using TNDM, by monitoring the dynamics during training and restarting/adjusting hyperparameters when necessary, our model can easily be used to find a biologically plausible set of latent dynamics that explain both behaviour and the neural population activity. We are happy to clarify this better in the paper.
>
> > The causal relationship is not always true for behavior and neural activity, considering feedback scenarios.
>
> This is true and something we have thought about extensively. To the best of our knowledge, no current latent variable model for neural populations has been extended to model feedback from the behaviour of interest (e.g. proprioception). For example, PSID assumes that neural activity gives rise to behaviour without any feedback. While we are interested in eventually extending TNDM to model this feedback loop, we believe that this is beyond the scope of our current submission as it would require significantly more challenging datasets and modelling assumptions.
>
> > "to learn low-dimensional dynamics" is ambiguous. In TNDM, the dynamical system is modeled by a high-D RNN as well as LFADS. Only the latent factors z are low-D.
>
> This is an excellent point and something we will correct in the updated manuscript. Indeed, the dimensionality of the dynamics in TNDM (and LFADS) is high as we use a high-D RNN to model the dynamics. As mentioned by the reviewer, it is actually the dimensionality of the subspace that gives rise to neural activity and behaviour (the latent factors z) which will be low due to the projection. Therefore, our model can be used to examine the number of latent variables (i.e. activity patterns) that are needed to characterize the population response and corresponding behaviour. As this is the primary goal when fitting LVMs to neural data (Cunningham et al. 2014), we compare all LVMs in this paper (TNDM, LFADS, and PSID) by the dimensionality of this subspace rather than the dimensionality of the dynamics.
>
> Citations:
> * Cunningham, John P., and M. Yu Byron. "Dimensionality reduction for large-scale neural recordings." Nature neuroscience 17.11 (2014): 1500-1509.
>
> > The power of SSM like LFADS is not only to recover the latent trajectories but also to learn the dynamical system (velocity/phase portrait/simulated trajectories). This should be also evaluated.
>
> We agree that SSMs can be probed to better understand the underlying dynamical system, however, we do not think this analysis would greatly improve our evaluation of TNDM in relation to PSID and to LFADS. Already, we evaluate TNDM on the neural reconstruction, behaviour reconstruction, and interpretability of the latent factors which we believe sufficiently shows the success of TNDM in relation to the other methods. More detailed analysis of the SSM could be used to address biological hypotheses, however, this is beyond the scope of our more technical paper. One experiment we would be happy to include in the supplement is the effect of removing one reach direction during training. We found that TNDM is still able to infer the initial conditions, dynamics, and behaviour for the neural activity associated with the held out reach direction despite never training on it.
>
> > It is expected that TNDM outperforms LFADS on behavior encode at small number factors, because of imperfect encoding of behavior variables and other internal dynamics.
>
> We fully agree with the reviewer that TNDM is expected to outperform LFADS on behaviour reconstruction. We compare TNDM to LFADS in our manuscript because LFADS is currently the best performing model for single trial neural activity, thus outperforming LFADS on behaviour reconstruction while simultaneously performing as well on neural reconstruction validates that our model is performing as expected. The most important quantitative comparison is between TNDM and PSID as both methods utilize behaviour as part of the generative process, therefore, the comparison is much more fair. Here, we show that TNDM far outperforms PSID in both behavioural and neural reconstruction while retaining a high-degree of interpretability.
>
> > It is also important to evaluate the disentanglement, e.g. the decoding performance of irrelevant dynamics.
>
> We actually perform this evaluation. In Supplement 5, we perform an ablation study of the disentanglement penalty where we show that the decoding performance of the irrelevant dynamics is significantly lower than the relevant dynamics (although not completely zero). This illustrates that our disentanglement penalty and model structure successfully encourages the behaviourally relevant dynamics to be separated from the behaviourally irrelevant dynamics.
>
> > PSID used Gaussian filtered spikes. This made the data and likelihood are both different from that of TNDM. Since TNDM is not limited to Poisson for spikes. It's more fair to fit TNDM to the same data and likelihood as PSID.
>
> We believe that the most fair comparison is between the best version of each model. Since TNDM can model both spikes and Gaussian filtered spikes while PSID can only model Gaussian filtered spikes (it fails when the smoothing is reduced), it should be fair to compare either version of TNDM to PSID. Not being able to model spikes directly is a limitation of PSID and its subspace identification approach which should be a factor in the evaluation. However, we would be happy to add this experiment if the reviewer still thinks it would be valuable.

---

> > ### Author Response · Authors · 2021-08-10
> > **Response to Reviewer wP6X Continued**
> >
> > > The authors spent a lot for the details of LFADS and PSID but not say much about how PSID separates dynamics which is a key point to the work.
> >
> > We agree that we should have added more details about how PSID separates the behaviourally relevant from the behaviourally irrelevant dynamics. The key insight of PSID is that the behaviorally relevant latent states lie in the intersection of the spaces spanned by past neural activity and future behavior. Therefore, in the first stage of PSID, it is possible to extract these states directly from the training data with an orthogonal projection of future behavior onto the past neural activity. Then, the second stage consists of an additional orthogonal projection from the residual neural activity onto past neural activity. In these two stages, PSID learns the two sets of dynamics separately such that the behaviourally relevant dynamics are unaffected by the behaviourally irrelevant dynamics. We would be happy to add a few sentences to the manuscript to clarify how PSID separates the two sets of dynamics.
> >
> > > In Eq(2), the encoder network is only a function of the spike  without behavior ? and why not?
> >
> > This was a deliberate modelling choice we made where we wanted the approximate variational posterior to only be a function of the neural data $X$. Importantly, not requiring paired data for the encoder provides an additional use for TNDM which is that it can be used to predict behaviour from neural activity.
> >
> > > It is unclear how the initial conditions Φr and Φi are parametrized in Eq(2).
> >
> > We realize that our notation may have been confusing. Φr1 and Φr2 are actually the weight matrices that map from the RNN encoder output to the mean and variance of the behaviourally relevant initial conditions. Φi1 and Φi2 are the weight matrices that map from the RNN encoder output to the mean and variance of the behaviourally irrelevant initial conditions.
> >
> > > Where are $\mathbf{g_0}_r$ and $\mathbf{g_0}_i$ defined? These bold symbols have different indexing order from their plain counterparts. Are they vectors?
> >
> > ${g_0}_r$ and ${g_0}_i$ are defined in Equation (1) as the initial conditions of the behaviorally relevant and behaviourally irrelevant dynamics. The “0” in the subscript indicates that they are the high-dimensional dynamics at time step zero (i.e. the initial conditions) and the “r” and the “i” in the subscript indicate that they are behaviourally relevant and irrelevant dynamics respectively. It does not appear these symbols are bold, however. We can add an additional sentence to clarify this notation.
> >
> > > How is the mean of the sample cross-correlations between $\mathbf{g_0}_r$ and $\mathbf{g_0}_i$ computed?
> >
> > We computed our disentanglement penalty by sampling from each initial condition distribution and then constructing a sample covariance matrix. We then took the mean of the squared sum of the elements of this covariance matrix. This penalty was used to reduce the linear correlation between the two initial condition distributions. We can add this detail to our manuscript.
> >
> > ### Minor Concerns:
> >
> > > Why to trim the trials to minimum? Varying length trials should not be a problem.
> >
> > This is a good point. We set the length of all trials to be the same to make the evaluation and plotting more convenient, however, there was no need to do this given that all the methods can handle varying length trials.
> >
> > > Why the covariance of behavior is fixed at the identity matrix?
> >
> > When using a Gaussian decoder (as we did for the behaviour), one can learn the variance with a diagonal covariance matrix. While powerful, this approach is prone to numerical instability (Rybkin et al. 2021). We chose to fix the variance to be constant (i.e. mean squared error) to overcome these instabilities. There may be better approaches to solving this numerical instability including learning a shared variance term, but we did not try this approach as mean squared error worked well.
> >
> > Citations:
> > * Rybkin, Oleh, Kostas Daniilidis, and Sergey Levine. "Simple and effective VAE training with calibrated decoders." International Conference on Machine Learning. PMLR, 2021.

---

> > > ### Comment · Reviewer_wP6X · 2021-08-22
> > > **Thanks for the authors response**
> > >
> > > Thanks for the response. After read the rebuttal, I have still some concerns.
> > >
> > > - I still think that the quality of learning the dynamics is important for state-space models, e.g. forecast or simulate meaningful trajectories. Inferring smoothing latent trajectories doesn't necessarily mean a good fit.
> > > - To show the method can be universally used, a variety of (synthetic) examples are needed. E.g. in the reaching task, disentanglement penalty is imposed on the distributions of the initial conditions. The conditions are categorical. Then what if the conditions are continuous, or even not explicit? It lacks a general guidelines.
> > > - Causal or anti-causal can be changed via design of the filter. I don't see much novelty here.

---

> > > > ### Author Response · Authors · 2021-08-23
> > > > **Clarifying potential misunderstanding of initial conditions**
> > > >
> > > > We thank the reviewer for getting back to us. We are glad that we were able to satisfactorily answer most of your original concerns and we hope to address your remaining concerns below:
> > > >
> > > > * Although we still believe our original evaluation was quite comprehensive (i.e. neural reconstruction, behaviour reconstruction, regression of the learned factors to the observed velocity, visualization, etc.), we understand that it would be valuable to provide additional evidence about the quality of the learned dynamics. We would be happy to add simulated trajectories and interpolations between inferred initial conditions in the final manuscript.
> > > > * We believe there is a misunderstanding of our generative model. **TNDM’s initial conditions are always assumed to be continuous and Gaussian (similar to Pandarinath et al., 2018.)**. This is detailed in Equation (2) of our paper where we define the variational posterior over the initial conditions as the product of two independent multivariate Gaussian distributions with diagonal covariances. Whether or not the 2D reaching behaviour has a discrete number of reach directions (i.e. reach direction is categorical) has nothing to do with the initial conditions of the dynamics which will be continuous and Gaussian in TNDM. Therefore, the disentanglement penalty should always be applied the same way and does not “lack any general guidelines”. **This is why we mentioned earlier in our rebuttal that TNDM is not tailored tightly to the reaching dataset and can be applied generically to any paired neural activity and observed behaviour.** As we now evaluate TNDM on synthetic data that is generated with these assumptions (i.e. the Lorenz system), we do not think that adding additional synthetic datasets will provide additional insights about our method. If the reviewer still feels as though we should add additional synthetic datasets, however, we would be happy to create more and add them to the final manuscript.
> > > > * While our behaviour decoder is simple to implement, it is conceptually novel in comparison to recent LVMs for modelling behaviour and neural activity (Sani et al. 2020). As can be seen in Supplement 1, changing our behaviour decoder to model arbitrary lags is essential for TNDM’s excellent performance on both neural reconstruction and behaviour decoding in comparison to the more constrained behaviour decoder (i.e. no lag) introduced by Sani et al. 2020. This is not only important because one cannot assume a fixed latency between activity and observed behaviour (e.g. inertia can differ), but also because it adds more flexibility for the latent factors to explain behaviour. For example, in Figure 4, we show that the behaviour decoder is integrating the behaviourally relevant factors for the reach data which implies that these factors are encoding the velocity of the 2D hand movement. We agree that the behaviour decoder could easily be changed to be anti-causal. In the anti-causal case, TNDM could also be quite interesting; for example, it could be used to extract the dynamics underlying some stimuli that the neural population is encoding. This flexibility also means that the performance of different decoders can be compared, for instance, to test hypotheses about the relationship between neural dynamics and externally recorded variables. Either way, our behaviour decoder is a novel conceptual contribution for LVMs of neural population activity and behaviour.
> > > >
> > > > Again, we thank the reviewer for their comprehensive review and follow-up questions. We hope that our response has sufficiently addressed any persistent concerns about our manuscript.

---

> > > > > ### Comment · Reviewer_wP6X · 2021-08-31
> > > > > **To authors**
> > > > >
> > > > > Thanks for the responses. Based on the clarification and promised more evaluation, I am willing to improve my initial score of the paper.

---

### Official Review · Reviewer_6ehA · 2021-07-16

**Rating:** 7
**Confidence:** 4

**Summary:**

In this paper authors provide a novel architecture for modelling neural dynamics with the aim of separating dynamics that are related to behaviour from the ones that are not. Using amortized inference, observed neural activities are used to set the initial state of two RNNs, one for relevant dynamics and one for irrelevant dynamics, which in turn predict behavioural and neural data. The results show that compared to baseline models the model is indeed able to predict behavioural responses, while maintaining prediction ability for the neural data.

**Ethical Concerns:**

I suggest presenting the ethics approval related to the used data.

**Limitations And Societal Impact:**

They are adequately discussed.

**Main Review:**

The paper is clearly written and the model, contributions, and the results are well explained. The model itself is also well designed for the purpose and potentially provides important contributions to the literature.

I have two concerns as explained below.

I am rather confused about why neural dynamics at time t are modelled to affect future behaviours at c > t directly (in addition to the behaviour at time t). Normally, I would assume that current neural dynamics at time t influence neural activity at time c > t, which in turn affect behaviour at time c. But the current setup implies that neural activities at time t had to directly affect behaviour at time c without mediation of neural responses at time c. It seems to imply that the RNN network was unable to capture the whole neural dynamics over time, which seems a serious limitation.

From the results, it's hard to evaluate whether irrelevant RNN is contributing at all to the prediction of neural activities; i.e., it could be the case that the whole neural dynamics are explained by the relevant RNN; is there any evidence supporting the role of relevan RNN in predicting neural activities? Similarly, how much is the contribution of the irrelevant RNN in predicting neural activities? Is there any evidence that not all the neural dynamics is explained by the irrelevant RNN?




**Time Spent Reviewing:**

2

---

> ### Author Response · Authors · 2021-08-10
> **Response to Reviewer 6ehA**
>
> ### General
>
> We thank Review 6ehA for the thoughtful review and useful comments. We are glad that the reviewer felt that the manuscript was clearly written and the contributions and results were well-explained. We also appreciate that the reviewer recognizes that our work makes an important contribution to the literature.
>
> ### Major Concerns
>
> > I am rather confused about why neural dynamics at time t are modelled to affect future behaviours at c > t directly (in addition to the behaviour at time t). Normally, I would assume that current neural dynamics at time t influence neural activity at time c > t, which in turn affect behaviour at time c. But the current setup implies that neural activities at time t had to directly affect behaviour at time c without mediation of neural responses at time c. It seems to imply that the RNN network was unable to capture the whole neural dynamics over time, which seems a serious limitation.
>
> When modelling neural population activity with a latent variable model, most state-space models have the Markovian property that the neural activity at time step $t$ only depends on the latent factor at time step $t$. Note that here, future neural activity depends on past activity because the latent factors are generated by a dynamical system. Importantly, the behaviour at time step $c$ may not have any relation to the neural activity at timestep $c$ because there is a potentially variable latency between brain and muscle activation, delays due to kinetics (e.g. inertia) and so on. *In fact, behaviour may be completely determined by the neural activity at time step $t < c$ which means that the behaviour should be generated by the latent factor at time step $t < c$ as well*. TNDM captures the latencies between the latent factors and behaviour with its fully casual behaviour decoder. As a result, the form of TNDM’s learned behaviour decoder can be interpretable; for instance, in Figure 4 we show that M1 activity best reflects movement velocity, not position or acceleration (i.e. each time point of the behaviour receives contributions from a broad time interval of preceding factor activity corresponding to a temporal integration of the factors). We hope this clarifies how behaviour decoding is done, and we are happy to explain this better in the manuscript.
>
> > From the results, it's hard to evaluate whether irrelevant RNN is contributing at all to the prediction of neural activities; i.e., it could be the case that the whole neural dynamics are explained by the relevant RNN; is there any evidence supporting the role of relevant RNN in predicting neural activities? Similarly, how much is the contribution of the irrelevant RNN in predicting neural activities? Is there any evidence that not all the neural dynamics is explained by the irrelevant RNN?
>
> The contributions of the relevant and irrelevant model components to neural activities can be seen in the magnitude of the weights from factors to log-rates (since this transformation is linear, they can be interpreted in terms of variance explained). We show this in Appendix 1 for a model where we enforce same-time decoding of neural activity and behaviour. In this case the relevant factors only contribute weakly to activity, hence the model does not learn a useful shared latent space (as an aside we note that we observe this behaviour in PSID). In contrast, the version of our model with a more flexible behaviour decoder (the one we evaluate throughout the paper) utilizes both latent spaces to reconstruct activity, as evidenced by the similar weight magnitudes.

---

> > ### Comment · Reviewer_6ehA · 2021-09-02
> > **Response to authors.**
> >
> > Thank you for your response. I am satisfied with the responses.

---

### Official Review · Reviewer_AfQj · 2021-07-17

**Rating:** 6
**Confidence:** 4

**Summary:**

This paper details a dimensionality reduction and dynamical modeling technique that partitions the neural variability into behaviorally-relevant and behaviorally-irrelevant factors. This idea was introduced and explored in preferential subspace identification (PSID), a reformulation of SSID (subspace identification), which models the latent dynamics as evolving linearly. This paper extends that to the nonlinear regime, using recurrent neural networks to model the evolution of the latent variables. The authors use the LFADS framework to model the temporal evolution, while partitioning the state into behaviorally-relevant and behaviorally-irrelevant factors, and not allowing for the behaviorally-irrelevant state to influence the behaviorally-relevant state.

**Limitations And Societal Impact:**

Yes

**Main Review:**

The authors may have severely misrepresented PSID. From Equation (4) of the Methods in Sani et al. (2021), PSID actually seems to allow the behaviorally-irrelevant states as having an effect on the behaviorally relevant states, and not in fact a block diagonal transition matrix A as the authors have mentioned (Question: is this why their implementation of PSID fails?). TNDM does not allow this influence to occur, which is a significant departure from the assumptions in PSID, and in fact, the assumptions for TNDM require a strong motivation. The assumption that the behaviorally-irrelevant dynamics are completely disjoint from the behaviorally-relevant dynamics implies that there are two completely independent processes occurring in the neural activity that require a separate set of dynamics to explain. On the contrary, papers such as Sussillo et al. (2015) and Russo et al. (2018) show that if one trains a network to produce behavioral outputs in the context of motor control, the network states can actually recapitulate neural activity, traditionally thought of as behaviorally-irrelevant, but that are in fact necessary to produce behavior. Moreover, the authors mention that PSID is hindered since it cannot incorporate lags between the neural activity and the behavior. However, the state z is a latent representation that can in fact accumulate past neural activity to lead to current behavior, and thus can incorporate lags between the two representations.

The authors find that the reconstruction of the firing rates using TNDM is dependent almost purely on the number of total factors that are used, regardless of how many of them are constrained to be behaviorally-relevant (Figure 2C). Please discuss. It would be interesting to include in Figure 2A,C the behavioral and neural predictions for # of behaviorally-irrelevant factors = 0.

Please explain why TNDM achieves a lower firing rate RMSE than LFADS with the same number of factors (Figure 2C) for # total factors > 4.

Please show the post-hoc behavior reconstruction using the LFADS factors in Figure 3 using the 4 factors. The behavior R^2 was high, so the reconstructions should be visually comparable to TNDM.

Please discuss and / or show the post-hoc partition of LFADS factors into behaviorally-relevant and behaviorally-irrelevant by decoding behavior from the factors (as done for PSID in this paper).

The paper would benefit from a better discussion of the structure and role of behaviorally-relevant vs. behaviorally-irrelevant dynamics.

The authors refer to the behaviorally-irrelevant factors and networks as simply ‘irrelevant’. I would encourage them to use the full term ‘behaviorally-irrelevant’ instead.

Originality: The paper is an extension of LFADS using ideas in PSID.
Quality: The paper could lead to interesting insights into the structure of behavioral vs. neural activity. However, in its current form, it is unclear what it is adding to the realm of different dimensionality reduction models of neural activity.
Clarity: The paper is written very clearly, and the figures are of high quality.
Significance: While the paper explores an interesting idea, it would be much improved with proper motivation of their model structure.

**Time Spent Reviewing:**

3

---

> ### Author Response · Authors · 2021-08-10
> **Response to Reviewer AfQj**
>
> ### General
>
> We thank the reviewer for taking the time to review our manuscript. It appears the strongest criticism in the review is about the modelling assumptions of TNDM in comparison to those of PSID. However, the criticism of TNDM’s assumptions appears to be predicated on the misunderstanding that TNDM is modelling the behaviourally relevant and irrelevant dynamics in a completely disjoint manner. *This is not the case.* According to the probabilistic model of TNDM, the true posterior over the latent variables $P({g_0}_i, {g_0}_r | x, y)$ cannot be factorized; that is, $z_i$ and $z_r$ (which are deterministic transforms of ${g_0}_i$ and ${g_0}_r$) are conditionally dependent given the observed data. We want to clarify that although we make a mean field variational approximation to the true posterior (i.e. the variational posterior is factorized), this does not change any modelling assumptions and, therefore, the two sets of dynamics maintain their statistical dependencies. The purpose of the disentanglement penalty was to reduce sharing of information/redundancy between the two sets of dynamics. Beyond this misunderstanding, it appears the strongest criticism in this review is not aimed at TNDM, but rather at the comparison of TNDM with PSID and some statements we made about PSID. We address the points below and will be happy to modify the paper accordingly. Yet we feel this is not a strong justification to recommend rejection, since PSID is not the focus of our paper, but merely serves as one of two state-of-the-art methods for comparison. As a linear method, the advantages of PSID are its simplicity and tractability, yet we think it is fair to say that non-linear methods should generally perform better since neural/behavioural dynamics are inherently nonlinear. This is not a criticism of PSID, but simply reflects the fact that each method has its individual strengths and weaknesses.
>
> ### Major/Minor Concerns
>
> > The authors may have severely misrepresented PSID. From Equation (4) of the Methods in Sani et al. (2021), PSID actually seems to allow the behaviorally-irrelevant states as having an effect on the behaviorally relevant states, and not in fact a block diagonal transition matrix A as the authors have mentioned (Question: is this why their implementation of PSID fails?).
>
> It is important to clarify that **we did not re-implement PSID**. We used the official Python implementation provided by the authors of PSID and we contacted them after submission of this paper to clarify if we had used their code correctly. There was a small issue with the preprocessing of the behaviour data, however, this did not change any of our conclusions as PSID still has much lower behaviour decoding than TNDM and the reconstruction quality of the neural activity was also still lower. We agree that our description of PSID’s state-transition matrix $A$ being block diagonal was incorrect. However, the behaviourally irrelevant states will not have an effect on the behaviourally relevant states in PSID. This is because the upper triangular portion of $A$ is zero in equation (4), therefore, the behaviorally-irrelevant states cannot have an effect on the behaviorally relevant states. This is also clear from PSID’s two stage training where the behaviourally relevant states are learned first and the behaviourally irrelevant states are learned from the residual neural activity.
>
> Interestingly, while inspecting PSID’s learned model parameters, we actually found evidence that *PSID does not learn latent dynamics that are informative about the neural activity for our dataset*. This is demonstrated by the fact that PSID’s learned state-transition matrix $A$ is approximately the identity with eigenvalues that have real value 1 and an insignificant complex component. This also means that relevant and irrelevant dynamics in PSID are effectively disjoint. To better understand if PSID’s state-transition matrix $A$, which is an identity matrix, still meaningfully characterizes the neural activity, we also trained PSID on time shuffled and trial shuffled neural data. In both cases, the state-transition matrix $A$ is approximately the identity matrix. These experiments suggest that the learned identity matrix is not informative about the neural activity for PSID on our dataset. Below is a table summarising the above results. In this table, we show the eigenvalues of the state-transition matrix A for PSID trained with 2 behaviourally relevant factor and 0 behaviourally irrelevant factors (the lack of behaviourally irrelevant factors should have no affect on the behaviourally relevant factors).
>
> |  Experiment |             Normal             |         Trial Shuffled        |      Time Shuffled     |
> |:----------------:|:------------------------------:|:-----------------------------:|:----------------------:|
> | Eigenvalues of A | [1.01 + 0.0016j, 1.01 - 0.0016j] | [1.01 + 0.0015j, 1.01 - 0.0015j] | [1.016, 1.018] |
>
> We postulate that the state-transition matrix $A$ is uninformative about the neural activity due to the nonlinear behaviour. The behaviour in our dataset is nonlinear across all trials due to the 8 different reach directions. To test if this caused PSID’s issues, we ran an experiment where we trained PSID multiple times with 1 to 8 reach directions. The state-transition matrix $A$ matrix quickly collapses to the identity matrix as the number of reach directions increases past 1. This implies that the multi-reaching behaviour is difficult for PSID as it can only capture linear dynamics. In fact, any ability of PSID to reconstruct the behaviour and neural activity actually heavily depends on the Kalman filter inference which can correct for these uninformative dynamics. Below is a table illustrating that the state-transition matrix $A$ matrix approaches the identity as the number of reach directions increases in the dataset. We again ran this experiment for PSID trained with 2 behaviourally relevant factors and 0 behaviourally irrelevant factors.
>
> | # reach directions |   1  |   2  |   3  |   4  |   5  |   6  |   7  |   8  |
> |:------------------:|:----:|:----:|:----:|:----:|:----:|:----:|:----:|:----:|
> | 2 norm of A matrix | 1.06 | 1.01 | 1.01 | 1.01 | 1.01 | 1.01 | 1.01 | 1.01 |
>
> We would be happy to include the code and text of this analysis in Supplement 6 to illustrate an example where linear dynamical methods such as PSID appear to be unable to model more complex datasets that may be better suited for nonlinear methods. We believe that this finding is of great importance to the neuroscience community and further highlights the strength of our nonlinear modelling approach.
>
> > TNDM does not allow this influence to occur, which is a significant departure from the assumptions in PSID, and in fact, the assumptions for TNDM require a strong motivation. The assumption that the behaviorally-irrelevant dynamics are completely disjoint from the behaviorally-relevant dynamics implies that there are two completely independent processes occurring in the neural activity that require a separate set of dynamics to explain. On the contrary, papers such as Sussillo et al. (2015) and Russo et al. (2018) show that if one trains a network to produce behavioral outputs in the context of motor control, the network states can actually recapitulate neural activity, traditionally thought of as behaviorally-irrelevant, but that are in fact necessary to produce behavior.
>
> This is a misunderstanding of TNDM and its modelling assumptions. It is incorrect to say that the behaviourally relevant/irrelevant dynamics are completely disjoint in TNDM. As mentioned above, the probabilistic model of TNDM implies that the true posterior over the latent variables $P({g_0}_i, {g_0}_r | x, y)$ cannot be factorized; that is, *$z_i$ and $z_r$, which are deterministic transforms of ${g_0}_i$ and ${g_0}_r$, respectively, are conditionally dependent given the observed data*. Although we make a mean field variational approximation to the true posterior (i.e. the variational posterior is factorized), this does not change any modelling assumptions and, therefore, the two sets of dynamics maintain their statistical dependencies.The disentanglement penalty was simply introduced to reduce sharing of information/redundancy between the two sets of dynamics as we demonstrate in Supplement 5. Also, it is also important to understand that while behaviourally relevant dynamics constitute all activity patterns required to reconstruct behaviour, behaviourally irrelevant dynamics may be completely unrelated (e.g. other behavioural or brain states), or even related to movement execution (e.g. dynamics that are associated with movement generally but are not task specific). Both TNDM and PSID offer a way to access these subspaces.

---

> > ### Author Response · Authors · 2021-08-10
> > **Response to Reviewer AfQj continued**
> >
> > > Moreover, the authors mention that PSID is hindered since it cannot incorporate lags between the neural activity and the behavior. However, the state z is a latent representation that can in fact accumulate past neural activity to lead to current behavior, and thus can incorporate lags between the two representations.
> >
> > While PSID can accumulate past neural activity to determine the current latent state, the main issue it has when modelling lags actually stems from the requirement that *PSID must simultaneously explain neural activity and behaviour at timestep $t$ with the latent factor z at time step $t$ with a Markov blanket of one*. This is a very strong constraint as it forces neural activity at timestep $t$ to contain information about the behaviour at timestep $t$ because they are generated by the same latent state with no skip connections. An example where this will cause issues is when the behaviour at time step $t$ is completely unrelated to the neural activity at time step $t$ because of latency between the neural activity and the behaviour (i.e. the time it takes between cortex activity and muscle activation). In this case, the latent factor at time step $t$ must make a tradeoff between explaining behaviour and explaining neural activity. TNDM loosens this rather implausible restriction by allowing for behaviour at time step $t$ to be directly modelled by previous latent states (and the latent state at timestep $t$), while still requiring that neural activity at timestep $t$ be explained by the latent state at timestep $t$. Therefore, TNDM’s dynamics can flexibly give rise to behaviour while still directly explaining the neural activity at each time step.
> >
> > > Please explain why TNDM achieves a lower firing rate RMSE than LFADS with the same number of factors (Figure 2C) for # total factors > 4.
> >
> > TNDM achieves a lower firing rate RMSE than LFADS with the same number of factors because the RMSE is computed for the average firing rate over all trials of the same movement direction. While TNDM and LFADS have a very similar Poisson likelihood on single trials, TNDM can better distinguish trials by movement direction since it is explicitly modelling behaviour, hence the firing rate prediction split by trial type is improved. We can add an explanation of this phenomenon to the paper.
> >
> > > The authors find that the reconstruction of the firing rates using TNDM is dependent almost purely on the number of total factors that are used, regardless of how many of them are constrained to be behaviorally-relevant (Figure 2C). Please discuss. It would be interesting to include in Figure 2A,C the behavioral and neural predictions for # of behaviorally-irrelevant factors = 0.
> >
> > This is an interesting question. We believe this phenomenon arises from the fact that the data set we used requires at least two behaviourally relevant factors to reconstruct behaviour, yet neural activity can always be captured by both behaviourally relevant and irrelevant factors. If behaviour reconstruction is poor because there are not enough behaviourally relevant factors, the behaviourally irrelevant space can compensate for this in the neural reconstruction. Conversely, if more behaviourally relevant factors are introduced, behaviour loss saturates quickly, and the behaviourally relevant factors can then be used to improve activity reconstruction. As long as sufficient variance in the activity is due to behaviour, this will lead to the observed behaviour that activity reconstruction depends only on the total number of factors. This raises an issue we did not discuss explicitly in the paper: investigating model performance using a range of behaviourally relevant/irrelevant factors can reveal if neural dynamics relating to behaviour are restricted to a low-dimensional manifold that corresponds to the dimensionality of the behaviour variables, or is even lower dimensional. For instance, in preliminary work we investigated motor cortex recordings together with 12 channel EMG (behaviour), and found that two behaviourally relevant factors are still sufficient for successful fits. If, on the other hand, neural dynamics are higher dimensional, we would observe a separation of irrelevant factors by behaviour type in the behaviourally irrelevant factors, which is not the case in the data set analysed in our paper. Taken together we think this is strong evidence that neuronal dynamics are constrained to low-dimensional non-linear manifolds. We are happy to add a comment to this effect.
> >
> > > Please show the post-hoc behavior reconstruction using the LFADS factors in Figure 3 using the 4 factors. The behavior $R^2$ was high, so the reconstructions should be visually comparable to TNDM.
> >
> > We would be happy to show this figure in the manuscript. As a reminder, when we evaluate LFADS on behaviour prediction, we have to perform an ex post facto regression from the extracted latent factors to the behaviour in the training set. The regression is linear and is from all time steps of the factors to all time steps of the behaviour. This means that the regression is more flexible than the decoder in TNDM which is also linear but constrained to be causal.
> >
> > > Please discuss and / or show the post-hoc partition of LFADS factors into behaviorally-relevant and behaviorally-irrelevant by decoding behavior from the factors (as done for PSID in this paper).
> >
> > Performing a post-hoc partition of the LFADS factors into behaviourally relevant and irrelevant factors is challenging as different sources of variability are mixed in the latent space. As can be seen in Figure 3, all 4 latent factors of LFADS encode information about the trial-specific behaviour. We take this into account when evaluating the behaviour prediction of LFADS by regressing all the LFADS factors to behaviour. We are not sure what you mean when you ask us to do for LFADS what we did for PSID. PSID allows us to set the size of the behaviourally relevant and irrelevant subspaces before training while LFADS does not.
> >
> > > The paper would benefit from a better discussion of the structure and role of behaviorally-relevant vs. behaviorally-irrelevant dynamics.
> >
> > The results suggest the latent dynamics are separable into components that are related to the specific type of movement (i.e. behaviourally relevant), and components that reflect a more generic activity pattern associated with movement (i.e. behaviourally irrelevant). In unpublished work, we found that the latter activity pattern makes a strong contribution to variance in the measured response, and, to some extent, masks out the specific movement-related activity patterns (these can be more prominent in directions of lower variance). This has potential implications for both our understanding of motor cortex dynamics and for effective decoding of movement intention. We would be happy to discuss these findings in more depth in the updated manuscript.
> >
> > > The authors refer to the behaviorally-irrelevant factors and networks as simply ‘irrelevant’. I would encourage them to use the full term ‘behaviorally-irrelevant’ instead.
> >
> > We can correct this in the updated manuscript.
> >
> > > Originality: The paper is an extension of LFADS using ideas in PSID. Quality: The paper could lead to interesting insights into the structure of behavioral vs. neural activity. However, in its current form, it is unclear what it is adding to the realm of different dimensionality reduction models of neural activity. Clarity: The paper is written very clearly, and the figures are of high quality. Significance: While the paper explores an interesting idea, it would be much improved with proper motivation of their model structure.
> >
> > We do not agree with this assessment of our work. We believe that the motivation and contribution of TNDM is actually quite clear: TNDM is the first nonlinear latent dynamical model that jointly models the observed behaviour and neural activity. Our proposed probabilistic graphical model is theoretically novel and our empirical results are state-of-the art as TNDM substantially outperforms the linear baseline, PSID, on both behavioral and neural reconstruction while offering a high-degree of interpretability in its latent dynamical factors.

---

> > > ### Comment · Reviewer_AfQj · 2021-08-31
> > > **Response to authors**
> > >
> > > Thank you for the detailed and comprehensive comments to all the reviews. They are very helpful and appreciated. I think some of the misunderstanding in both mine and Reviewer wP6X reviews may stem from clarity of the writing; I am hoping the authors will clarify the model structure in the revision and provide more detail regarding the assumptions. Based on this and the provided rebuttal, I am willing to improve my initial score of the paper.

---

### Author Response · Authors · 2021-08-10
**General Response**

We thank the reviewers for their detailed feedback and thoughtful questions/comments. At a high-level, our response to the reviewers is focused on clarifying the novelty and contribution of TNDM, addressing concerns with our comparison to PSID, and describing additional experiments we ran to validate TNDM.

We want to clarify that the main contribution of TNDM is in its novel probabilistic graphical model for disentangling the non-linear behaviourally relevant and irrelevant dynamics that underlie behaviour and neural activity. Not only is our model theoretically novel, it also far outperforms the current state-of-the-art baseline, PSID, in both neural and behavioural reconstruction. We also believe that TNDM’s prediction about M1 activity during reach, that the behaviourally relevant dynamics represent information about movement velocity, is an important contribution and of interest for the neuroscience community.

Additionally, to address Reviewer wP6X’s concerns about the lack of diverse datasets in the manuscript, we have now run TNDM and LFADS on a synthetic dataset (generated with a Lorenz system) and evaluated their ability to recover the latent dynamical factors. We show that on this dataset, TNDM matches (and potentially surpasses) the performance of LFADS on recovering the factors. This, together with further analysis, shows that TNDM works as intended to recover interpretable latent dynamics from neural recordings. For more details on this experiment and the results, please see our response to Reviewer wP6X. We would be happy to add a new supplement to our manuscript that details the dataset, hyperparameters, and results for this experiment.

---

### Decision · Program_Chairs · 2021-09-27

**Decision:**

Accept (Poster)

**Comment:**

This paper proposes a new sequential VAE that simultaneously models behavioral and neural time series. The reviewers agree that it provides meaningful contribution to neural data analysis. The discussion among the reviewers and the authors was very constructive and productive, and the overall writing of the final version should reflect the nuances raised through the process and results from the additional experiments.